# GaussianCube: A Structured and Explicit Radiance Representation for 3D Generative Modeling

**Bowen Zhang**[1*] **Yiji Cheng**[2*] **Jiaolong Yang**[3†] **Chunyu Wang**[3†]

**Feng Zhao**[1‡] **Yansong Tang**[2] **Dong Chen**[3‡] **Baining Guo**[3]

[1]University of Science and Technology of China [2]Tsinghua University [3]Microsoft Research Asia

## Abstract

We introduce a radiance representation that is both structured and fully explicit and thus greatly facilitates 3D generative modeling. Existing radiance representations either require an implicit feature decoder, which significantly degrades the modeling power of the representation, or are spatially unstructured, making them difficult to integrate with mainstream 3D diffusion methods. We derive GaussianCube by first using a novel densification-constrained Gaussian fitting algorithm, which yields high-accuracy fitting using a fixed number of free Gaussians, and then rearranging these Gaussians into a predefined voxel grid via Optimal Transport. Since GaussianCube is a structured grid representation, it allows us to use standard 3D U-Net as our backbone in diffusion modeling without elaborate designs. More importantly, the high-accuracy fitting of the Gaussians allows us to achieve a high-quality representation with orders of magnitude fewer parameters than previous structured representations for comparable quality, ranging from one to two orders of magnitude. The compactness of GaussianCube greatly eases the difficulty of 3D generative modeling. Extensive experiments conducted on unconditional and class-conditioned object generation, digital avatar creation, and text-to-3D synthesis all show that our model achieves state-of-the-art generation results both qualitatively and quantitatively, underscoring the potential of GaussianCube as a highly accurate and versatile radiance representation for 3D generative modeling. Project page: https://gaussiancube.github.io/.

## 1 Introduction

The field of 3D generation [59, 39, 5, 57, 49, 8, 19, 11, 61, 76] has witnessed remarkable growth, driven by advancements in generative modeling [25, 20, 41, 17, 72, 29]. Most of the prior works in this domain leverage variants of Neural Radiance Field (NeRF) [38, 8, 57, 40] as their underlying 3D representations, which typically consist of an explicit structured proxy representation and an implicit feature decoder. However, such hybrid NeRF variants exhibit degraded representation power, particularly when used for generative modeling where a single implicit feature decoder is shared across all objects. Additionally, the high computational complexity of volumetric rendering leads to both slow rendering speed and extensive memory costs.

Recently, the emergence of 3D Gaussian Splatting (GS) [30] has enabled improved reconstruction quality and real-time rendering capabilities [69, 36, 63, 35]. The fully explicit nature of 3DGS eliminates the need for a shared implicit decoder, providing another key advantage over NeRFs. Although 3DGS has been widely studied in scene reconstruction tasks, its spatially unstructured nature presents a significant challenge when applied to mainstream generative modeling frameworks.

---

*Interns at Microsoft Research Asia. †Equal advising. ‡Corresponding authors.

38th Conference on Neural Information Processing Systems (NeurIPS 2024).

| Representation | Spatially-structured | Fully-explicit | Real-time Rendering | Rel. Parameters↓ |
|---|---|---|---|---|
| Instant-NGP [38] | ✗ | ✗ | ✗ | 26.63× |
| Neural Voxels [57] | ✓ | ✗ | ✗ | 145.9× |
| Triplane [8] | ✓ | ✗ | ✗ | 13.7× |
| Gaussian Splatting [30] | ✗ | ✓ | ✓ | 4.0× |
| **Our GaussianCube** | ✓ | ✓ | ✓ | **1.0×** |

Table 1: Comparison with previous 3D representations with respect to spatial structure, explicitness, real-time rendering capability, and relative parameter count (Rel. Parameters) for representations of comparable quality.

To overcome these barriers, we introduce GaussianCube – an innovative radiance representation that is both structured and fully explicit, with strong fitting capabilities (see Table 1 for comparisons with prior works). The proposed approach first ensures high-accuracy fitting with a predefined number of free Gaussians, and subsequently organizes these Gaussians into a structured voxel grid. Such an explicit grid-based structure permits the seamless application of standard 3D convolutional architectures, such as U-Net, thereby eliminating the need for complex, specialized network designs [77, 59] that are often necessary with unstructured or implicitly decoded representations.

Structuring 3D Gaussians without sacrificing fitting quality is not a trivial task. A naive starting point would be obtaining a fixed number of Gaussians by omitting the densification and pruning steps in GS. However, such simplification fails to lead the Gaussians close to the object surfaces and results in significant quality degradation. In contrast, we propose a *densification-constrained fitting* strategy, which retains the original pruning process yet constrains the number of Gaussians that perform densification, ensuring the total does not exceed a predefined maximum $N^3$. For the subsequent structuralization, we allocate the Gaussians across an $N \times N \times N$ voxel grid using *Optimal Transport (OT)*. Consequently, our fitted Gaussians are systematically arranged within the voxel grid, with each voxel containing the features of a Gaussian. The proposed OT-based structuralization achieves maximal spatial correspondence, characterized by minimal total transport distances, while preserving the expressive power of 3DGS.

The structured nature of GaussianCube enables us to perform efficient 3D diffusion [25] modeling for the following three reasons: 1) It allows the use of standard 3D U-Net as our backbone for diffusion modeling without elaborate designs. 2) The spatial coherence of GaussianCube permits the use of standard 3D convolutions to capture the correlations among neighboring Gaussians, facilitating efficient feature extraction. 3) GaussianCube enables high-quality fitting with orders of magnitude fewer parameters than prior structured representations of similar quality. Since recent works [32, 3] have demonstrated diffusion models' struggle in handling high-dimensional distributions, the compactness of GaussianCube significantly reduces the modeling difficulty of the generative framework.

We conduct comprehensive experiments to verify the efficacy of our approach. The model's capability for unconditional and class-conditioned generation is evaluated on the ShapeNet [9] and OmniObject3D [64] datasets. Both the quantitative and qualitative comparisons indicate that our model surpasses all previous methods. We also perform digital avatar generation on a synthetic avatar dataset [62]. Our model is capable of producing high-fidelity 3D avatars conditioned on single portrait images, excelling beyond prior art in both identity preservation and detail creation. Additionally, we assess our model's capacity for the challenging text-to-3D creation task on Objaverse [14]. Our model demonstrates competitive performance both quantitatively and qualitatively, producing results consistent with the given text prompts in just 2.3 seconds. All experiments show the strong capabilities of our GaussianCube and suggest its potential as a powerful and versatile 3D representation for a variety of applications. Some generated samples of our method are presented in Figure 1.

## 2 Related Work

**Radiance field representation.** Radiance fields model ray interactions with scene surfaces and can be in either implicit or explicit forms. Early works of neural radiance fields (NeRFs) [38, 74, 43, 1, 45] are often in an implicit form, which represents scenes without defining geometry. These works optimize a continuous scene representation using volumetric ray-marching that leads to extremely high computational costs. Recent works introduce the use of explicit proxy representation [8, 26, 18, 51, 40, 68] followed by an implicit feature decoder to enable faster rendering. Recently, the 3D Gaussian Splatting methods [30, 69, 63, 13, 31, 10, 71] utilize 3D Gaussians as their underlying representation and offer impressive reconstruction quality. The fully explicit representation also provides real-time rendering speed. However, the 3D Gaussians are unstructured representation, and

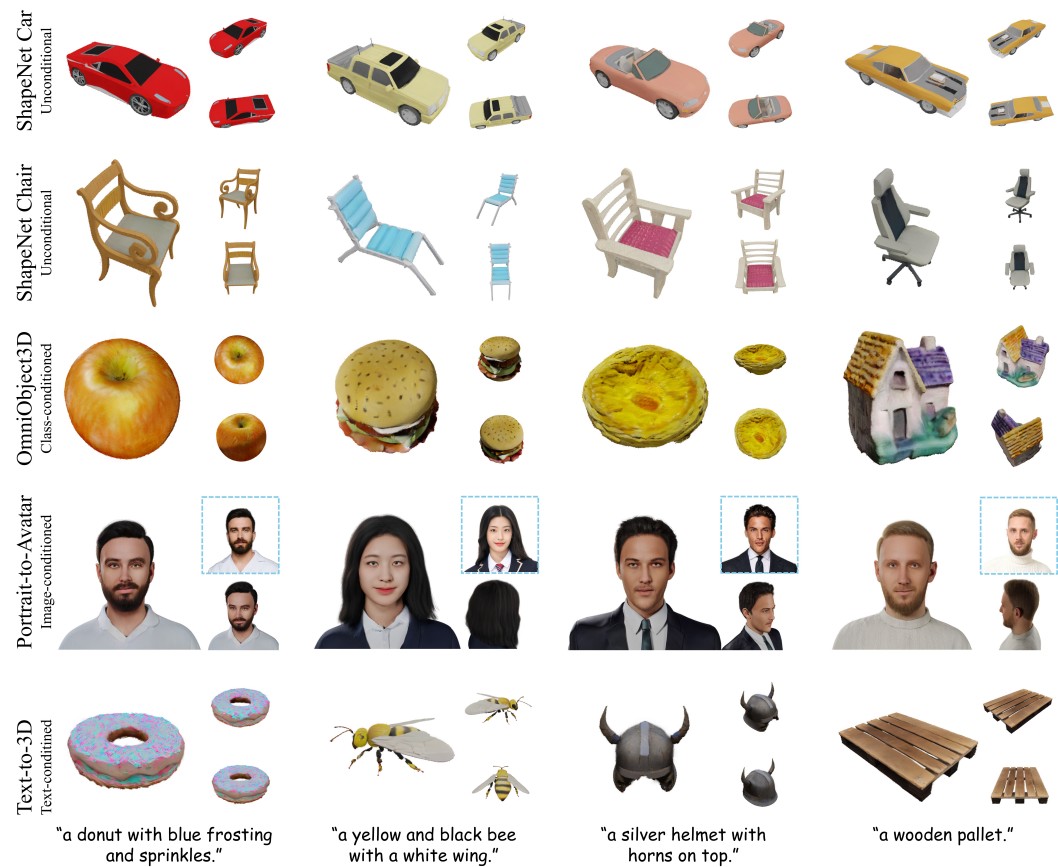

Figure 1: Our diffusion model is able to create diverse objects with complex geometry and rich texture details (top three rows). Our method also supports creating high-fidelity digital avatars (the forth row) conditioned on single portrait images (visualized in dashed boxes) and high-quality 3D assets given text prompts (the fifth row).

require per-scene optimization to achieve photo-realistic quality. In contrast, our work proposes a structured representation termed GaussianCube for 3D generative tasks.

**3D generation.** Previous works of SDS-based optimization [44, 55, 67, 52, 12, 53, 70, 56] distill 2D diffusion priors [47] to a 3D representation with the score functions, but these works are notably time-intensive, often taking several minutes to hours. While 3D-aware GANs [8, 19, 7, 21, 42, 16, 66] facilitate view-dependent image generation from single-image collections, they struggle to capture the complexity of diverse objects with intricate geometric variations [65]. Although recent works [59, 39, 22, 57, 49, 73] have utilized diffusion models with structured proxy representations for 3D generation, the use of a shared implicit feature decoder across different assets restricts expressiveness and the computational demands of NeRF hinder efficient training. In contrast, we introduce a structured and fully explicit radiance representation for 3D generative modeling, building upon 3DGS [30]. A concurrent work of [23] includes elaborate designs to form the Gaussians into volumetric representation during fitting, yet does not thoroughly address global correspondence. In contrast, our approach only restricts the total count of Gaussians while allowing freedom in their spatial distribution during the fitting. We then organize these Gaussians into a voxel grid using Optimal Transport, which yields a spatially coherent arrangement with minimal global offset cost, effectively easing the difficulty of generative modeling.

## 3 Method

Following prior works, our framework comprises two primary stages as shown in Figure 2: representation construction and diffusion modeling. In representation construction phase, we first apply a densification-constrained 3DGS fitting algorithm for each object to obtain a constant number of Gaussians. These Gaussians are then organized into the proposed spatially structured GaussianCube

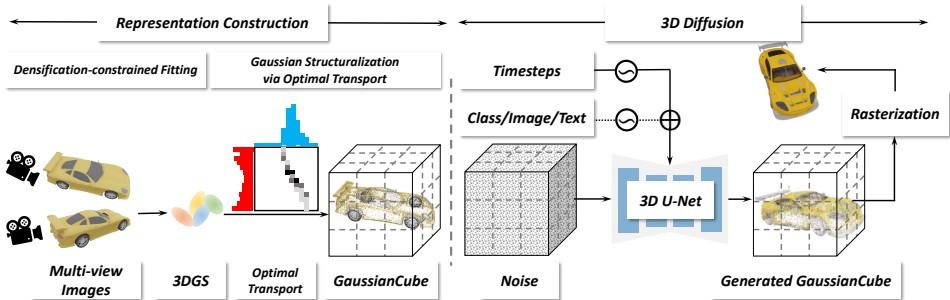

Figure 2: **Overall framework.** Our framework comprises two main stages of representation construction and 3D diffusion. In the representation construction stage, given multi-view renderings of a 3D asset, we perform *densification-constrained fitting* to obtain 3D Gaussians with constant numbers. Subsequently, the Gaussians are structured into GaussianCube via *Optimal Transport*. In the 3D diffusion stage, our *3D diffusion model* is trained to generate GaussianCube from Gaussian noise.

via Optimal Transport between the positions of Gaussians and centers of a predefined voxel grid. For diffusion modeling, we train a 3D diffusion model to learn the distribution of GaussianCubes. We will detail our designs for each stage subsequently.

## 3.1 Representation Construction

We build our GaussianCube upon 3DGS, an explicit representation that offers impressive fitting quality and real-time rendering speed. However, it fails to yield Gaussians of fixed length since the adaptive density control during GS fitting can lead to a varying number of Gaussians for different objects. Furthermore, the lack of a predetermined spatial ordering for Gaussians leads to a disorganized spatial structure. These aspects pose significant challenges to 3D generative modeling. To overcome these obstacles, we first introduce our densification-constrained fitting strategy to obtain a fixed number of free Gaussians. Then, we systematically arrange the resulting Gaussians within a predefined voxel grid via Optimal Transport, thereby achieving a structured and explicit radiance representation.

Formally, a 3D asset is represented by a collection of 3D Gaussians as introduced in Gaussian Splatting [30]. The geometry of the $i$-th 3D Gaussian $\boldsymbol{g}_i$ is given by

$$\boldsymbol{g}_i(\boldsymbol{x}) = \exp\left(-\frac{1}{2}\left(\boldsymbol{x}-\boldsymbol{\mu}_i\right)^\top \boldsymbol{\Sigma}_i^{-1}\left(\boldsymbol{x}-\boldsymbol{\mu}_i\right)\right),\tag{1}$$

where $\boldsymbol{\mu}_i \in \mathbb{R}^3$ is the center of the Gaussian and $\boldsymbol{\Sigma}_i \in \mathbb{R}^{3\times 3}$ is the covariance matrix defining the shape and size, which can be decomposed into a quaternion $\boldsymbol{q}_i \in \mathbb{R}^4$ and a vector $\boldsymbol{s}_i \in \mathbb{R}^3$ for rotation and scaling, respectively. Moreover, each Gaussian $\boldsymbol{g}_i$ have an opacity value $\alpha_i \in \mathbb{R}$ and a color feature $\boldsymbol{c}_i \in \mathbb{R}^3$ for rendering. Combining them together, the $C$-channel feature vector $\boldsymbol{\theta}_i = \{\boldsymbol{\mu}_i, \boldsymbol{s}_i, \boldsymbol{q}_i, \alpha_i, \boldsymbol{c}_i\} \in \mathbb{R}^C$ fully characterizes the Gaussian $\boldsymbol{g}_i$.

**Densification-constrained fitting**. Our approach begins with the aim of maintaining a constant number of Gaussians $\boldsymbol{g} \in \mathbb{R}^{N_{\max} \times C}$ across different objects during the fitting. A simplistic approach might involve omitting the densification and pruning in the original GS. However, we argue that such simplifications significantly harm the fitting quality, with empirical evidence shown in Table 6. Instead, we propose to retain the pruning process while imposing a new constraint on the densification phase as shown in Figure 3 (a). The fitting process encompasses several distinct stages: 1) Densification Detection: Assuming the current iteration includes $N_c$ Gaussians, we identify densification candidates by selecting those with view-space position gradient magnitudes exceeding a predefined threshold $\tau$. We denote the number of candidates as $N_d$. 2) Candidate sampling: To prevent exceeding the predefined maximum of $N_{\max}$ Gaussians, we select $\min\left(N_{\max} - N_c, N_d\right)$ Gaussians with the largest view-space positional gradients from the candidates for densification. 3) Densification: We modify the densification approach by alternating between cloning and splitting actions into separate steps. 4) Pruning Detection and Pruning: We identify and remove the Gaussians with $\alpha$ less than a small threshold $\epsilon$. After completing the fitting process, we pad Gaussians with $\alpha = 0$ to reach the target count of $N_{\max}$ without affecting the rendering results. Benefiting from our proposed strategy, we

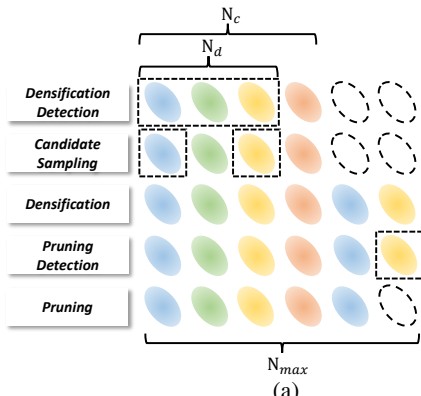 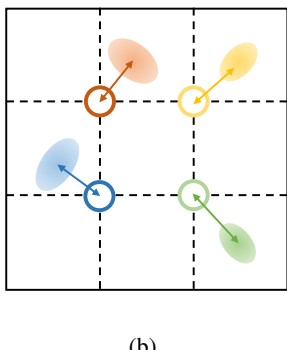

| (a) | (b) |

Figure 3: **Illustration of representation construction.** First, we perform densification-constrained fitting to yield a fixed number of Gaussians, as shown in (a). We then employ Optimal Transport to organize the resultant Gaussians into a voxel grid. A 2D illustration of this process is presented in (b).

attain a high-quality representation with orders of magnitude fewer parameters compared to existing works of similar quality, which significantly reduces the modeling difficulty for the diffusion models.

**Gaussian structuralization via Optimal Transport**. To further organize the obtained Gaussians into a spatially structured representation for 3D generative modeling, we propose to map the Gaussians to a predefined structured voxel grid $\boldsymbol{v} \in \mathbb{R}^{N_v \times N_v \times N_v \times C}$ where $N_v = \sqrt[3]{N_{\max}}$. Intuitively, we aim to "move" each Gaussian into a voxel while preserving their geometric relations as much as possible. While naive approaches such as nearest neighbor transport fall short in conserving these relations due to disregard for global arrangement with evidence shown in Figure 10, we formulate this as an Optimal Transport (OT) problem [58, 4] between the Gaussians' spatial positions $\{\boldsymbol{\mu}_i, i = 1, \ldots, N_{\max}\}$ and the voxel grid centers $\{\boldsymbol{x}_j, j = 1, \ldots, N_{\max}\}$. Let $\mathbf{D}$ be a distance matrix with $\mathbf{D}_{ij}$ being the moving distance between $\boldsymbol{\mu}_i$ and $\boldsymbol{x}_j$, i.e., $\mathbf{D}_{ij} = \|\boldsymbol{\mu}_i - \boldsymbol{x}_j\|^2$. The transport plan is represented by a binary matrix $\mathbf{T} \in \mathbb{R}^{N_{\max} \times N_{\max}}$, and the optimal transport plan is given by:

$$
\begin{aligned}
\underset{\mathbf{T}}{\text{minimize}} \quad & \sum_{i=1}^{N_{\max}} \sum_{j=1}^{N_{\max}} \mathbf{T}_{ij} \mathbf{D}_{ij} \\
\text{subject to} \quad & \sum_{j=1}^{N_{\max}} \mathbf{T}_{ij} = 1 \quad \forall i \in \{1, \ldots, N_{\max}\} \\
& \sum_{i=1}^{N_{\max}} \mathbf{T}_{ij} = 1 \quad \forall j \in \{1, \ldots, N_{\max}\} \\
& \mathbf{T}_{ij} \in \{0, 1\} \quad \forall (i, j) \in \{1, \ldots, N_{\max}\} \times \{1, \ldots, N_{\max}\}.
\end{aligned}
\tag{2}
$$

The solution is a bijective transport plan $\mathbf{T}^*$ that minimizes the total transport distances. We employ the Jonker-Volgenant algorithm [27] to solve the OT problem. We provide a 2D illustration in Figure 3 (b). We organize the Gaussians according to the solutions, with the $j$-th voxel encapsulating the feature vector of the corresponding Gaussian $\boldsymbol{\theta}_k = \{\boldsymbol{\mu}_k - \boldsymbol{x}_j, \boldsymbol{s}_k, \boldsymbol{q}_k, \alpha_k, \boldsymbol{c}_k\} \in \mathbb{R}^C$, where $k$ is determined by the optimal transport plan (*i.e.*, $\mathbf{T}^*_{kj} = 1$). Note that we replace the original Gaussian positions with offsets of the current voxel center to reduce the solution space for diffusion models. As a result, our fitted Gaussians are systematically arranged within a voxel grid $\boldsymbol{v}$ and preserve the spatial correspondence of neighboring Gaussians, which further facilitates generative modeling.

### 3.2 3D Diffusion on GaussianCube

We now introduce our 3D diffusion model incorporated with the proposed expressive, efficient and spatially structured representation. After organizing the fitted Gaussians $\boldsymbol{g}$ into GaussianCube $\boldsymbol{y}$ for each object, we aim to model the distribution of GaussianCube, *i.e.*, $p(\boldsymbol{y})$.

Formally, the generation procedure can be formulated into the inversion of a discrete-time Markov forward process. During the forward phase, we gradually add noise to $\boldsymbol{y}_0 \sim p(\boldsymbol{y})$ and obtain a sequence of increasingly noisy samples $\{\boldsymbol{y}_t | t \in [0, T]\}$ according to $\boldsymbol{y}_t := \alpha_t \boldsymbol{y}_0 + \sigma_t \boldsymbol{\epsilon}$, where $\boldsymbol{\epsilon} \in \mathcal{N}(\boldsymbol{0}, \boldsymbol{I})$ represents the added Gaussian noise, and $\alpha_t, \sigma_t$ constitute the noise schedule. As a result, $\boldsymbol{y}_T$ will finally reach isotropic Gaussian noise after sufficient destruction steps. By reversing the above process, we are able to perform the generation process by gradually denoise the sample starting from pure Gaussian noise $\boldsymbol{y}_T \sim \mathcal{N}(\boldsymbol{0}, \boldsymbol{I})$ until reaching $\boldsymbol{y}_0$. Our diffusion model is trained to denoise $\boldsymbol{y}_t$ into $\boldsymbol{y}_0$ for each timestep $t$, facilitating both unconditional and conditional generation.

Table 2: Comparison with prior 3D representations of spatial structure, fitting quality, relative fitting speed (Rel. Speed) and parameter sizes on ShapeNet Car. * denotes that the implicit feature decoder is shared across different objects. All methods are evaluated at 30K iterations.

| Representation | Spatially-structured | PSNR↑ | LPIPS↓ | SSIM↑ | Rel. Speed↑ | Params (M)↓ |
|---|---|---|---|---|---|---|
| Instant-NGP | ✗ | 33.98 | 0.0386 | 0.9809 | 1× | 12.25 |
| Gaussian Splatting | ✗ | **35.32** | **0.0303** | **0.9874** | 2.58× | 1.84 |
| Voxels | ✓ | 31.78 | 0.0676 | 0.9664 | 0.15× | 67.12 |
| Voxels* | ✓ | 30.25 | 0.0926 | 0.9541 | 0.15× | 67.12 |
| Triplane | ✓ | 32.61 | 0.0611 | 0.9709 | 1.05× | 6.30 |
| Triplane* | ✓ | 31.39 | 0.0759 | 0.9635 | 1.05× | 6.30 |
| **Our GaussianCube** | ✓ | 34.94 | 0.0347 | 0.9863 | **3.33×** | **0.46** |

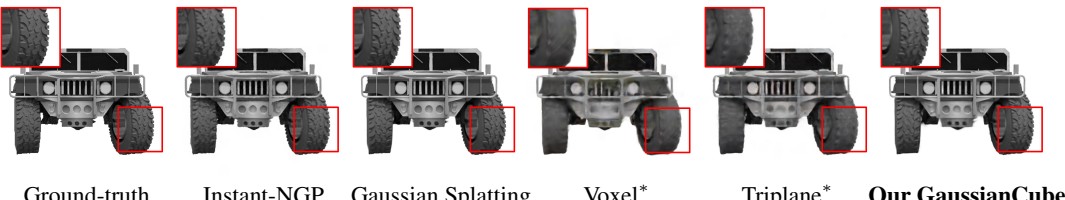

Ground-truth    Instant-NGP    Gaussian Splatting    Voxel*    Triplane*    **Our GaussianCube**

Figure 4: Qualitative results of object fitting.

**Model architecture.** Thanks to the spatially structured organization of the proposed GaussianCube, standard 3D convolution is sufficient to effectively extract and aggregate the features of neighboring Gaussians without elaborate designs. We leverage the standard U-Net network for diffusion [41, 17] and simply replace the original 2D operators including convolution, attention, upsampling and downsampling with their 3D counterparts.

**Conditioning mechanism.** Our model supports a variety of condition signals to control the generation process. When performing class-conditioned diffusion modeling, we employ adaptive group normalization (AdaGN) [17] to inject the class labels into our model. For image-conditioned digital avatar creation, we leverage a pretrained vision transformer [6] to encode the conditional image into a sequence of feature tokens. We subsequently adopt cross-attention to make the model learn the correspondence between 3D activations and 2D image feature tokens following [5]. We also leverage cross-attention as our condition mechanism when creating 3D objects from text, similar to previous text-to-image diffusion models [47].

**Training objective.** In our 3D diffusion training, we parameterize our model $\hat{\boldsymbol{y}}_\theta$ to predict the noise-free input $\boldsymbol{y}_0$ using:

$$\mathcal{L}_{\text{simple}} = \mathbb{E}_{t, \boldsymbol{y}_0, \boldsymbol{\epsilon}} \left[ \|\hat{\boldsymbol{y}}_\theta \left( \alpha_t \boldsymbol{y}_0 + \sigma_t \boldsymbol{\epsilon}, t, \boldsymbol{c}_{\text{cls}} \right) - \boldsymbol{y}_0 \|_2^2 \right], \tag{3}$$

where the condition signal $\boldsymbol{c}_{\text{cls}}$ is only needed when training conditional diffusion models. We additionally impose image-level supervision to improve the rendering quality of generated GaussianCube, which has been demonstrated to effectively enhance the perceptual details in previous works [59, 39]. Specifically, we penalize the discrepancy between the rasterized images $I_{\text{pred}}^t$ of the model prediction at timestep $t$ and the ground-truth images $I_{\text{gt}}$ using:

$$\mathcal{L}_{\text{image}} = \mathbb{E}_{I_{\text{pred}}^t} \left( \sum_l \left\| \Psi^l \left( I_{\text{pred}}^t \right) - \Psi^l \left( I_{\text{gt}} \right) \right\|_2^2 \right) + \mathbb{E}_{I_{\text{pred}}^t} \left( \left\| I_{\text{pred}}^t - I_{\text{gt}} \right\|_2 \right), \tag{4}$$

where $\Psi^l$ is the multi-resolution feature extracted using the pre-trained VGG [50]. Benefiting from the efficiency of both rendering speed and memory costs from GS [30], we are able to perform fast training with high-resolution renderings. Our overall training loss can be formulated as:

$$\mathcal{L} = \mathcal{L}_{\text{simple}} + \lambda \mathcal{L}_{\text{image}}, \tag{5}$$

where $\lambda$ is a balancing weight.

Table 3: Quantitative results of unconditional generation on ShapeNet Car and Chair [9] and class-conditioned generation on OmniObject3D [64].

| Method | ShapeNet Car | | ShapeNet Chair | | OmniObject3D | |
|---|---|---|---|---|---|---|
| | FID-50K↓ | KID-50K(‰)↓ | FID-50K↓ | KID-50K(‰)↓ | FID-50K↓ | KID-50K(‰)↓ |
| EG3D | 30.48 | 20.42 | 27.98 | 16.01 | - | - |
| GET3D | 17.15 | 9.58 | 19.24 | 10.95 | - | - |
| DiffTF | 51.88 | 41.10 | 47.08 | 31.29 | 46.06 | 22.86 |
| **Ours** | **13.01** | **8.46** | **15.99** | **9.95** | **11.62** | **2.78** |

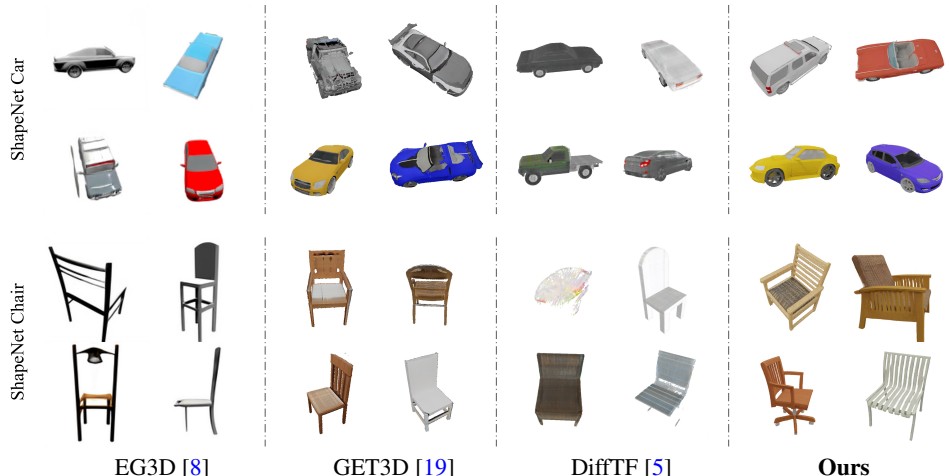

EG3D [8]  GET3D [19]  DiffTF [5]  **Ours**

Figure 5: Qualitative comparison of unconditional 3D generation on ShapeNet Car and Chair datasets. Our model is capable of generating results of complex geometry with rich details.

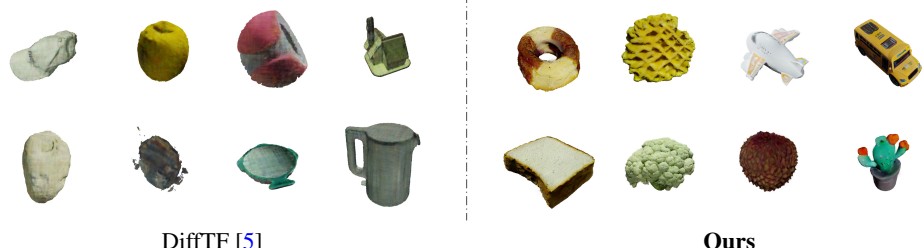

DiffTF [5]        **Ours**

Figure 6: Qualitative comparison of class-conditioned 3D generation on large-vocabulary OmniObject3D [64]. Our model is able to handle diverse distribution with semantically accurate results.

## 4 Experiments

### 4.1 Dataset and Metrics

To measure the expressiveness and efficiency of various 3D representations, we fit 100 objects in ShapeNet Car [9] using each representation and report the PSNR, LPIPS [75] and Structural Similarity Index Measure (SSIM) metrics when synthesizing novel views. Furthermore, we conduct experiments of single-category unconditional generation on ShapeNet [9] Car and Chair, and class-conditioned generation on real-world scanned dataset OmniObject3D [64]. We compute the FID [24] and KID [2] scores between 50K generated renderings and 50K ground-truth renderings. For image-conditioned digital avatar generation, we utilize the synthetic avatar dataset [62], which comprises highly-detailed 3D avatars created by synthetic pipeline. We assess the generation quality of 5K rendering from 500 test avatars and additionally include cosine similarity of identity embedding [15] (CSIM) to measure the ID preservation. The experiments of text-to-3D generation are performed on the large-scale challenging Objaverse dataset [14]. We numerically evaluate the text alignment quality using CLIP score [46] of 300 test prompts. All images are rendered with $512 \times 512$ resolution. For more details of data, please refer to Appendix A.1.

Table 4: Quantitative results of digital avatar creation conditioned on single portrait image.

| Method | PSNR↑ | LPIPS↓ | SSIM↑ | CSIM↑ | FID-5K↓ | KID-5K(‰)↓ |
|---|---|---|---|---|---|---|
| Rodin w/o 2D SR | 18.80 | 0.2842 | 0.7439 | 0.6594 | 32.07 | 24.78 |
| Rodin | 18.59 | 0.2821 | 0.7373 | 0.6466 | 20.02 | 9.24 |
| **Ours** | **21.87** | **0.1768** | **0.7703** | **0.7821** | **8.32** | **2.67** |

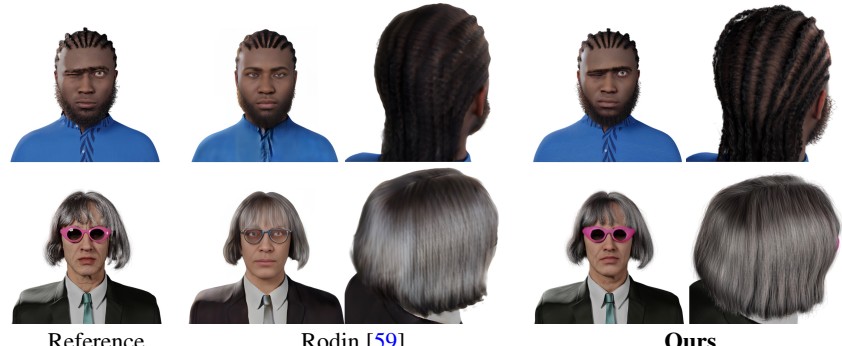

| Reference | Rodin [59] | **Ours** |

Figure 7: Qualitative comparison of 3D avatar creation conditioned on single frontal portraits.

## 4.2 Implementation Details

For GaussianCube construction, we set $N_{max}$ to 32,768 and $C$ to 14 across all datasets. We perform the proposed densification-constrained fitting for 30K iterations, which requires approximately 2.67 minutes on a single V100 GPU for each object. After OT-based structuralization, we obtain $32 \times 32 \times 32 \times 14$ GaussianCube for each object. The OT-based structuralization takes around 2 minutes per object on an AMD EPYC 7763v CPU. For the 3D diffusion model, we adopt the ADM U-Net network [41, 17]. We perform full attention at the resolution of $8^3$ and $4^3$ within the network. The timesteps of diffusion models are set to $1,000$ and we train the models using the cosine noise schedule [41] with loss weight $\lambda$ set to 10. We deploy 16 Tesla V100 GPUs for the ShapeNet Car, ShapeNet Chair, OmniObject3D, and Synthetic Avatar datasets, whereas 32 Tesla V100 GPUs are used for training on the Objaverse dataset. It takes about one week to train our model on ShapeNet Car, ShapeNet Chair, and OmniObject3D, and approximately two weeks for the Synthetic Avatar and Objaverse datasets. For more training details, please refer to Appendix A.1.

## 4.3 Main Results

**3D fitting.** We first evaluate our representation capability of object fitting against previous NeRF-based representations including Voxels [57] and Triplane [8], which are widely adopted in previous 3D generation works [8, 59, 5, 39, 57]. We set the representation size of Voxels and Triplane to $128 \times 128 \times 128 \times 32$ and $256 \times 256 \times 32$ respectively for comparable fitting quality. We also include Instant-NGP [40] and original Gaussian Splatting [30] for reference despite their unsuitability for generative modeling due to their unstructured spatial nature. As shown in Table 2, our GaussianCube outperforms all NeRF-based representations among all metrics. Figure 3 illustrates that GaussianCube can faithfully reconstruct geometry details and intricate textures. Moreover, we achieve such high-quality fitting with orders of magnitude fewer parameters than previous structured representation due to the densification-constrained fitting, showcasing our compactness. Notably, the shared implicit feature decoder in the multi-object fitting of NeRF-based methods leads to significant decreases in quality compared to single-object fitting as evidenced in Table 2. While the fully explicit nature of GS results in no quality gap between single and multiple object fitting.

**Single-category unconditional generation.** For unconditional generation, we compare our method with the state-of-the-art 3D generation works including 3D-aware GANs [8, 19] and Triplane diffusion models [5]. As shown in Table 3, our method surpasses all prior works in terms of both FID and KID scores and sets new records. We provide visual comparisons in Figure 5, where EG3D and DiffTF tend to generate blurry results with poor geometry, and GET3D fails to provide satisfactory textures. In contrast, our method yields high-fidelity results with authentic geometry and sharp textures.

Table 5: Quantitative results of text-to-3D creation. Inference time is measured on a single A100 GPU. While Shape-E, LGM achieve comparable CLIP scores as ours, they either utilize millions of training data or leverage 2D diffusion prior.

| | DreamGaussian | VolumeDiffusion | Shap-E | LGM | Ours |
|---|---|---|---|---|---|
| CLIP Score↑ | 26.38 | 24.41 | 30.52 | 30.06 | **30.56** |
| Inference Time (s)↓ | ∼ 120 | 4.95 | 4.42 | **1.55** | 2.30 |

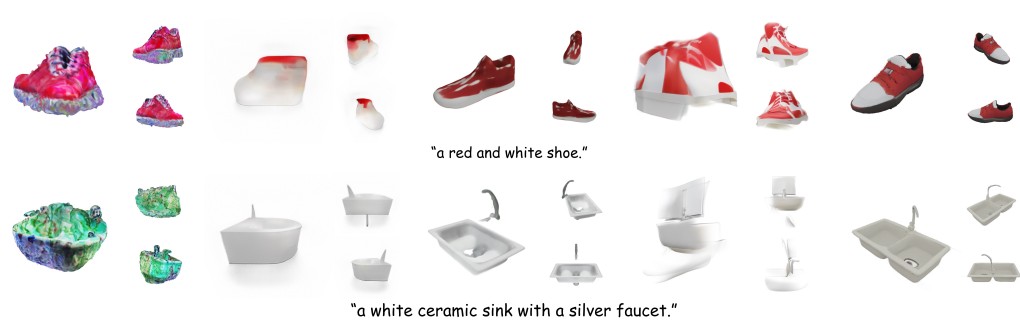

"a red and white shoe."

"a white ceramic sink with a silver faucet."

DreamGaussian [53]   VolumeDiffusion [57]   Shap-E [28]   LGM [54]   **Ours**

Figure 8: Qualitative comparison of text-to-3D generation on Objaverse [14]. Our model is able to generate high-quality samples according to the given text prompts.

**Large-vocabulary class-conditioned generation.** We also compare class-conditioned generation with DiffTF [5] on more diverse and challenging OmniObject3D [64] dataset. We achieve significantly better FID and KID scores than DiffTF as shown in Table 3. Visual comparisons in Figure 6 reveal that DiffTF often struggles to create intricate geometry and detailed textures, whereas our method is able to generate objects with complex geometry and realistic textures.

**Image-conditioned avatar generation.** For 3D avatar generation conditioned on a single reference image, we compare our method with state-of-the-art Triplane diffusion models, Rodin [47]. Our model surpasses Rodin among all evaluated metrics as shown in Table 4. Although Rodin utilizes a 2D refiner [60] to boost the visual quality of facial areas, which significantly compromises 3D consistency. Our model still outperforms it by direct real-3D generation. Results in Figure 7 demonstrate that our model faithfully preserves the identity, expression and accessories of the references with rich details, while Rodin struggles to provide satisfactory results even using 2D refinement.

**Text-to-3D generation.** We compare text-to-3D generation with prior arts including diffusion models [28, 57], optimization-based method [53] and feed-forward method [54]. Our model achieves competitive text-3D alignment results as shown in Table 5. The visual comparison in Figure 8 shows that our model is able to create high-quality samples aligning with text prompts in just 2.3 seconds. DreamGaussian tends to create over-saturated results and suffers from Janus problem. VolumeDiffusion produces unsatisfactory textures with poor text alignment. Shap-E can produce semantically accurate results but struggles to generate complex geometry. LGM reconstructs 3D Gaussians from multi-view images generated by text-conditioned multi-view diffusion pipeline [48], whereas the inconsistency [54] of the generated multi-views often results in inaccurate geometric reconstruction.

### 4.4 Ablation Study

We first examine the key factors in representation construction on ShapeNet Car. To spatially structure the Gaussians, a simplistic approach would be anchoring the positions of Gaussians to a predefined voxel grid while omitting densification and pruning, which leads to severe failure when fitting the objects as shown in Figure 9. Even by introducing learnable offsets to the voxel grid, the results still lack details. We observe the offsets are typically too small to effectively lead the Gaussians close to the object surfaces, which indicates the importance of densification in the fitting process. Instead, GaussianCube can capture both complex geometry and intricate details as shown in Figure 9. The numerical comparison in Table 6 also demonstrates the superior fitting quality of GaussianCube.

Table 6: Quantitative ablation of both representation fitting and generation quality on ShapeNet Car.

| Method | Densify & Prune | Representation Fitting | | | Generation | |
| --- | --- | --- | --- | --- | --- | --- |
| | | PSNR↑ | LPIPS↓ | SSIM↑ | FID-50K↓ | KID-50K(‰)↓ |
| A. Voxel grid w/o offset | ✗ | 25.87 | 0.1228 | 0.9217 | - | - |
| B. Voxel grid w/ offset | ✗ | 30.18 | 0.0780 | 0.9628 | 40.52 | 24.35 |
| C. Ours w/o OT | ✓ | **34.94** | **0.0346** | **0.9863** | 21.41 | 14.37 |
| **D. Ours** | ✓ | **34.94** | **0.0346** | **0.9863** | **13.01** | **8.46** |

Ground-truth    Table 6 A.    Table 6 B.    **Table 6 D. (Ours)**

Figure 9: Qualitative ablation of representation fitting.

**OT (Ours)**    Nearest Neighbor    Table 6 B.    Table 6 C.    **Table 6 D. (Ours)**

(a)    (b)

Figure 10: Visual ablation of the Gaussian organization methods and 3D generation. For visualization of Gaussian structuralization in (a), we map the coordinates of the corresponding voxel of each Gaussians to RGB values to visualize the organization. Our OT-based solution also results in the best generation quality shown in (b).

We also evaluate how the representation affects 3D generative modeling on ShapeNet Car as shown in Table 6 and Figure 10. Limited by the poor fitting quality, performing diffusion modeling on voxel grid with learnable offsets leads to blurry generation results as shown in Figure 10. To validate the importance of organizing Gaussians via Optimal Transport (OT), we compare with the organization based on nearest neighbor transport. We linearly map each Gaussian's corresponding coordinates of voxel to RGB color to visualize different organizations. As shown in Figure 10 (a), our proposed OT approach yields smooth color transitions, indicating that our method successfully preserves the spatial correspondence. However, nearest neighbor results in abrupt color transitions due to their disregard for global structure. Both the quantitative results in Table 6 and visual comparisons Figure 10 indicate that our globally structured arrangement facilitates generative modeling by alleviating its complexity, successfully leading to superior generation quality.

## 5    Conclusion

We have presented GaussianCube, a structured and explicit radiance representation crafted for 3D generative models. We begin by fitting each 3D object with a constant number of Gaussians using our proposed densification-constrained fitting algorithm. We further organize the obtained Gaussians into a spatially structured representation by solving the Optimal Transport between the positions of Gaussians and the predefined voxel grid. The proposed GaussianCube is spatially structured, allowing to use standard 3D U-Net for diffusion modeling without elaborate designs. Moreover, GaussianCube can achieve high-quality fitting using much fewer parameters compared to prior works of similar quality, which further eases the difficulty of generative modeling. Our 3D diffusion models equipped with GaussianCube achieve state-of-the-art generation quality on the evaluated datasets, underscoring its potential of GaussianCube as a versatile and powerful radiance representation for 3D generation.

**Acknowledgments:** This work was supported in part by the Anhui Provincial Natural Science Foundation under Grant 2108085UD12. We acknowledge the support of GPU cluster built by MCC Lab of Information Science and Technology Institution, USTC. We also thank anonymous reviewers for their valuable comments.

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

# A Appendix

## A.1 Additional Implementation Details

**Dataset preparation.** We conduct experiments on ShapeNet Car [9], ShapeNet Chair [9], OmniObject3D [64], Synthetic Avatar [62] and Objaverse [14] datasets. For each dataset, we report the total number of objects used for training, the number of views rendered per object for GaussianCube fitting and the distribution of camera poses used for rendering in Table 7. For the Objaverse dataset, we excluded low-quality objects, such as those without textures or with defective reconstructions following [57]. We also report the object bounding box $b$ in the world coordinate system of each dataset in Table 7, which is used to construct the predefined voxel grid within $[-b, b]^3$ during OT-based Gaussian structuralization.

**Representation construction.** We set $N_{\max}$ to 32768 and $C$ to 14 omitting the view-dependent spherical harmonics. This simplification appears to have a negligible impact on object fitting while concurrently reducing the data dimension, thereby alleviating the difficulty of diffusion modeling. During our densification-constrained fitting procedure, we primarily follow the hyper-parameters in original Gaussian Splatting [30]. For OT-based Gaussian structuralization, we adopt an approximate solution for the OT problem due to the $O\left(N_{\max}^3\right)$ time complexity of Jonker-Volgenant algorithm [27]. This is achieved by dividing the positions of the Gaussians and the voxel grid into four sorted segments and then applying the Jonker-Volgenant solver to each segment individually. We empirically find this approximation successfully strikes a balance between computational efficiency and spatial structure preservation. The proposed densification-constrained fitting takes around 2.67 minutes for each object of 30K iterations and the OT-based voxelization takes around 2 minutes which can be run on CPU in parallel.

**3D Diffusion.** To train the 3D diffusion model, we initially compute the instance-wise statistics of mean $\bar{\boldsymbol{\mu}} \in \mathbb{R}^{N_v \times N_v \times N_v \times C}$ and standard deviation $\bar{\boldsymbol{\sigma}} \in \mathbb{R}^{N_v \times N_v \times N_v \times C}$, from the GaussianCubes of each training dataset respectively. These statistical measures are then utilized to normalize the training data. For our 3D diffusion model architecture, we adopt the ADM-UNet from [17] and replace the convolution, upsampling, downsampling and attention operations with 3D implementations. We train our model using AdamW optimizer [33], and apply exponential moving average (EMA) with a rate of 0.9999 during training. We clamp the prediction of opacity $\alpha$ to $[0, 1)$ and clamp the minimum value of predicted scaling $s$ to 0 to ensure validity. For unconditional generation on ShapeNet, we train the model with a base learning rate $5e - 5$ for 850K iterations and then decay the learning rate to $5e - 6$ for another 150K iterations. For 3D digital avatar creation from a single portrait image, we adopt the pretrained DINO ViT-B/16 [6] to encode the $512 \times 512$ conditional images into $1025 \times 768$ conditional feature tokens. For text-to-3D creation, we take CLIP-L/14 [46] to encode the text prompts into $77 \times 768$ conditional feature tokens. We provide more detailed configurations of the model architectures, diffusion training and inference for each dataset in Table 8.

**Implementation of Gaussian organization visualization in Figure 10 (a).** For the $i$-th Gaussian, we obtain its corresponding voxel grid centers $\boldsymbol{x}_k \in \mathbb{R}^3$ according to Optimal Transport plan $\mathbf{T}^*$ (*i.e.*, $\mathbf{T}_{ik}^* = 1$) as illustrated in Section 3.1. To visualize the coordinates of $\boldsymbol{x}_k$, we map them to RGB color $\boldsymbol{C}_k \in \mathbb{R}^3$ using:

$$\boldsymbol{C}_k = \frac{(\boldsymbol{x}_k + \boldsymbol{b})}{2\boldsymbol{b}} \times \mathbf{255}, \tag{6}$$

where $\boldsymbol{b}$ is the bounding box in the world coordinate system. The resultant point cloud like visualizations are shown in Figure 10 (a), where smooth color transitions indicate coherent spatial correspondence preservation.

## A.2 Additional Ablation Study and Analysis

**Ablation of $N_{\mathbf{max}}$ in densification-constrained fitting.** We conduct experiments to evaluate how $N_{\max}$ affects fitting on ShapeNet Car. The results in Table 9 indicate that there is a clear trend where increasing $N_{\max}$ leads to improved fitting accuracy. However, a larger $N_{\max}$ also incurs higher computational costs during diffusion training. Therefore, we set $N_{\max}$ to 32,768 to strike a balance between high-quality fitting and computational efficiency.

Table 7: Details of each dataset.

| Dataset | # Objects | # Views per object | Rotation Angle | Elevation Angle | Bounding Box |
|---|---|---|---|---|---|
| ShapeNet Car | 7,462 | 150 | $[0, 2\pi]$ | $[\frac{1}{6}\pi, \frac{1}{2}\pi]$ | 0.45 |
| ShapeNet Chair | 6,775 | 150 | $[0, 2\pi]$ | $[\frac{1}{6}\pi, \frac{1}{2}\pi]$ | 0.35 |
| OmniObject3D | 5,795 | 100 | $[0, 2\pi]$ | $[0, \frac{1}{2}\pi]$ | 1.0 |
| Synthetic Avatar | 98,000 | 300 | $[0, 2\pi]$ | $[\frac{1}{6}\pi, \frac{2}{3}\pi]$ | 40.0 |
| Objaverse | 125,653 | 150 | $[0, 2\pi]$ | $[0, \frac{2}{3}\pi]$ | 0.5 |

Table 8: Detailed configuration of model architecture, diffusion training and inference on each dataset.

| | ShapeNet Car | ShapeNet Car | OmniObject3D | Synthetic Avatar | Objaverse |
|---|---|---|---|---|---|
| Diffusion Steps | 1,000 | 1,000 | 1,000 | 1,000 | 1,000 |
| Noise Schedule | Cosine | Cosine | Cosine | Cosine | Cosine |
| NFEs | 300 | 300 | 300 | 250 | 44 |
| Inference Time (s) | 10.06 | 10.06 | 10.06 | 13.80 | 2.30 |
| Inference Sampler | DPM-solver [34] | DPM-solver [34] | DPM-solver [34] | DPM-solver [34] | DPM-solver [34] |
| DPM-solver Order | 3 | 3 | 3 | 2 | 2 |
| DPM-solver Mode | Multi-step | Multi-step | Multi-step | Multi-step | Adaptive |
| CFG Scale | - | - | 2.0 | 1.3 | 3.5 |
| Model Size | 82M | 82M | 82M | 339M | 339M |
| Channels | 64 | 64 | 64 | 128 | 128 |
| Channel Mult. | (1,2,3,4) | (1,2,3,4) | (1,2,3,4) | (1,2,3,4) | (1,2,3,4) |
| Num. Res. Blocks | 3 | 3 | 3 | 3 | 3 |
| Attn Resolutions | (8, 4) | (8, 4) | (8, 4) | (8, 4) | (8, 4) |
| Num. Head Channels | 64 | 64 | 64 | 64 | 64 |
| Dropout | 0 | 0 | 0 | 0 | 0 |
| Scale Shift Norm | True | True | True | True | True |
| Training Steps | 1,000K | 1,000K | 700K | 1,200K | 1,800K |
| Training GPUs | 16 | 16 | 16 | 16 | 32 |
| Batch Size | 128 | 128 | 128 | 128 | 256 |
| Base Lr | $5e-5$ | $5e-5$ | $5e-5$ | $5e-5$ | $5e-5$ |
| Lr Decay Steps | 850K | 850K | - | - | - |

**Ablation of classifier-free guidance in class-conditioned generation.** We study how classifier-free guidance (CFG) impacts our generation quality when inference class-conditioned diffusion models. We report the FID and KID metrics in Table 10 under different CFG scales.

**Visualization of intermediate results in the denoising process.** During inference, our model starts from Gaussian noise and progressively denoises to yield the high-quality GaussianCube. We present visualizations of the intermediate renderings $y_t$ at various timesteps $t \in [0, T]$ throughout the denoising process, offering a detailed insight into the GaussianCube diffusion procedure. As illustrated in Figure 11, our model first establishes the global structure and then incrementally enhances the details, which is similar to previous 3D diffusion models [59, 49].

**Nearest neighbors analysis.** We perform nearest neighbor search of some unconditionally generated samples in the paper according to the similarity of pretrained CLIP [46] features. The results in Figure 12 demonstrate that our model is capable of generating novel geometry and textures rather than simply memorizing the training data.

**Distribution visualization of offset from voxel grids of fitted GaussianCubes.** We visualize the offset distribution of 1K randomly selected GaussianCubes from each experimental dataset in Figure 14. We observe that most distributions exhibit a bell curve, similar to a normal distribution. However, the Digital Synthetic Avatar dataset presents a more uniform distribution with multiple peaks. We believe these distributions offer valuable insights into how well the fitted 3D Gaussians align with voxel grid centers. Bell-shaped distributions akin to a normal distribution, such as in the ShapeNet Car and Chair datasets, suggest a strong initial alignment and lower complexity. On the other hand, broader distributions (e.g., the Digital Synthetic Avatar dataset) indicate a higher level of detail (for instance, hair) and a greater need for adjustments during organization.

## A.3 Additional Visual Results

For 3D avatar generation, while trained on synthetic dataset, our model is capable of generalizing to in-the-wild portrait input. We provide more visual comparison of 3D avatar creation conditioned on in-

Table 9: Quantitative ablation of $N_{\max}$ in densification-constrained fitting. We set $N_{\max}$ to 32,768 in this paper.

| $N_{\max}$ | $N_v$ | **PSNR↑** | **LPIPS↓** | **SSIM↑** |
|---|---|---|---|---|
| 4096 | 16 | 32.56 | 0.0547 | 0.9765 |
| 13824 | 24 | 34.32 | 0.0396 | 0.9842 |
| 32768 | 32 | 34.94 | 0.0347 | 0.9863 |
| 110592 | 48 | 35.29 | 0.0307 | 0.9874 |
| 262144 | 64 | 35.34 | 0.0301 | 0.9875 |

Table 10: Quantitative ablation of CFG scale in the class-conditioned generation of OmniObject3D [64].

| Scale | w/o CFG | 1.3 | 1.5 | 2.0 | 3.0 | 6.0 |
|---|---|---|---|---|---|---|
| **FID-50K↓** | 13.39 | 12.07 | 11.72 | **11.62** | 12.99 | 32.80 |
| **KID-50K(‰)↓** | 4.01 | 3.12 | 3.00 | **2.78** | 3.17 | 14.36 |

the-wild portraits with Rodin [59] in Figure 15. We also include additional comparison conditioned on synthetic input from our test in Figure 16. Our model can faithfully retain the identity of the reference portrait and is able to provide high-fidelity results with rich details, *e.g.* hair, glasses and clothing. Although utilizing a pretrained 2D super-resolution module which significantly compromises 3D consistency, Rodin struggles to follow the conditional images and fails to produce detailed textures in non-facial areas *e.g.* clothing and hair.

We include additional qualitative comparison and generated samples of text-to-3D generation in Figure 17 and Figure 18 respectively. Our model yields samples with better visual quality, and is capable of handling challenging prompts. The results in Figure 19 show the generation diversity of our results given the same text prompt. Our model is also capable of performing text-guided editing of generated objects by leveraging SDEdit [37] as depicted in Figure 20, demonstrating the promise of achieving controllable 3D generation.

We provide more generated samples of unconditional and class-conditioned generation in Figure 21, Figure 22 and Figure 23. The additional results demonstrate the strong capability of our model to create high-quality 3D assets with complex geometry and intricate textures.

Furthermore, we also provide an additional video in supplementary material, which intuitively illustrates our approach and visualizes the generated results.

## A.4 Limitations

While GaussianCube represents a substantial step forward in developing an ideal representation for 3D content generation, it still has some limitations. Specifically, although the GaussianCube construction procedure is considerably more rapid than that of NeRF-based methods and can be executed in parallel, it still requires approximately 5 minutes to construct each object. This presents a challenge for scaling up training on extensive 3D datasets. In future work, we plan to investigate more time-efficient methods for GaussianCube construction. Additionally, akin to prior 2D diffusion models, our text-to-3D diffusion model encounters difficulties in presenting the specified number of objects within prompts as shown in Figure 13. To address this, we will look into enhancing the precision and controllability of 3D generation in the future.

## A.5 Broader Impacts

The proposed GaussianCube enables high-quality 3D asset fitting with few parameters, which significantly simplifies the challenges of 3D generative modeling. Our diffusion model is capable of generating high-quality 3D assets of complex geometry and intricate textures while also accommodating a variety of conditional signals to steer the creating procedure. The strong capability of GaussianCube suggests its potential to serve as a versatile 3D representation for a variety of applications in future 3D research endeavors.

Like all generative models, particular caution is required when dealing with sensitive tasks involving human representations. Our avatar creation model is trained exclusively on a synthetic dataset [62] composed of large-scale 3D digital avatars which are generated through a graphics pipeline. We conceptualize digital avatars as analogous to those created by specialized 3D artists, rather than

photorealistic human images. This strategy in selecting training data mitigates privacy and copyright issues that might arise from utilizing real human photo collections. Nevertheless, it is crucial to acknowledge that avatars generated by our model from real-world imagery could still be misused for spreading disinformation. As such, we advocate implementing rigorous safeguards and promoting responsible use of our technology other related ones to mitigate such risks.

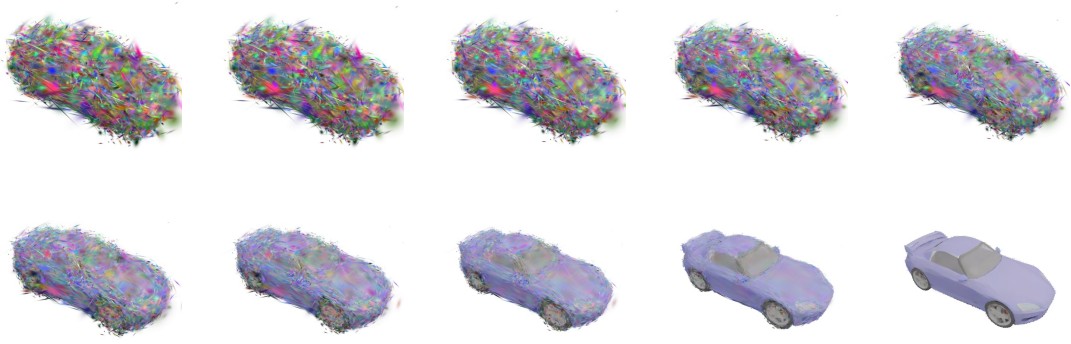

Figure 11: Visualization of generation results in intermediate diffusion timesteps.

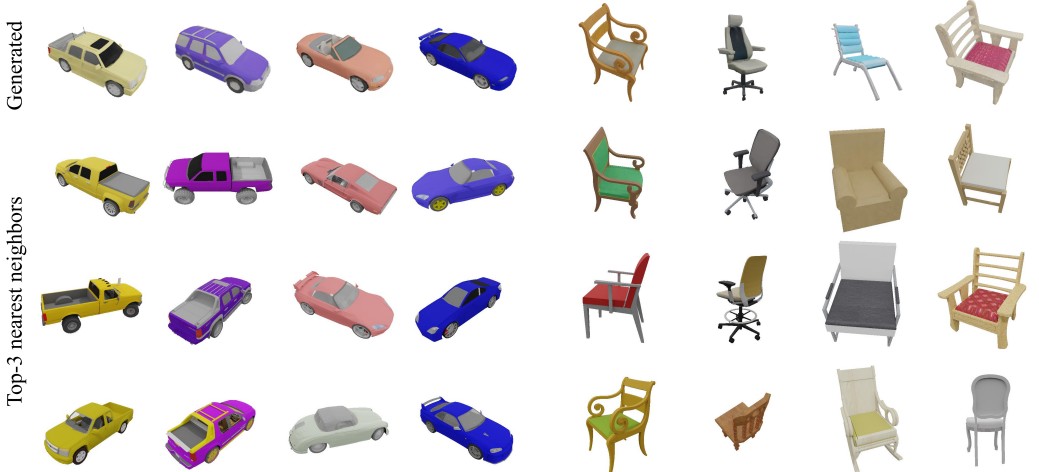

Figure 12: Visualization of nearest neighbor search on ShapeNet Car and Chair.

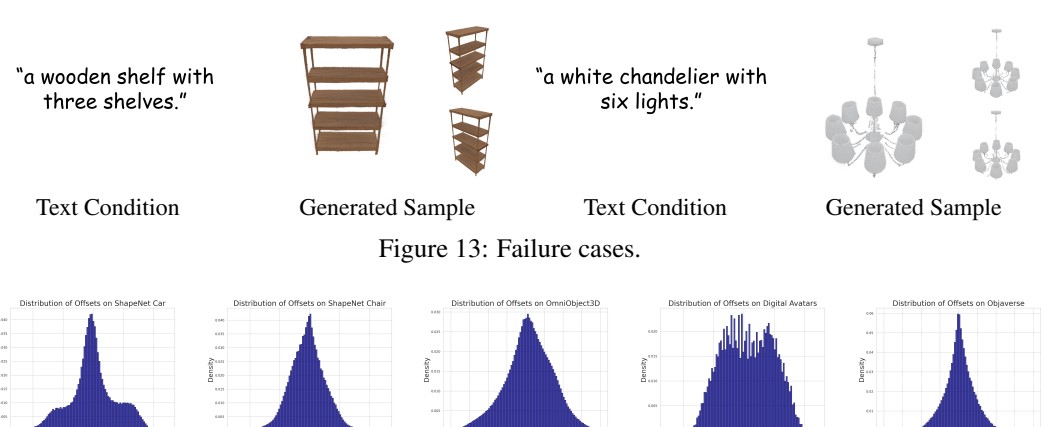

Figure 13: Failure cases.

Figure 14: Distribution of offsets from voxel centers in a random selection of 1K GaussianCubes on each experimental dataset.

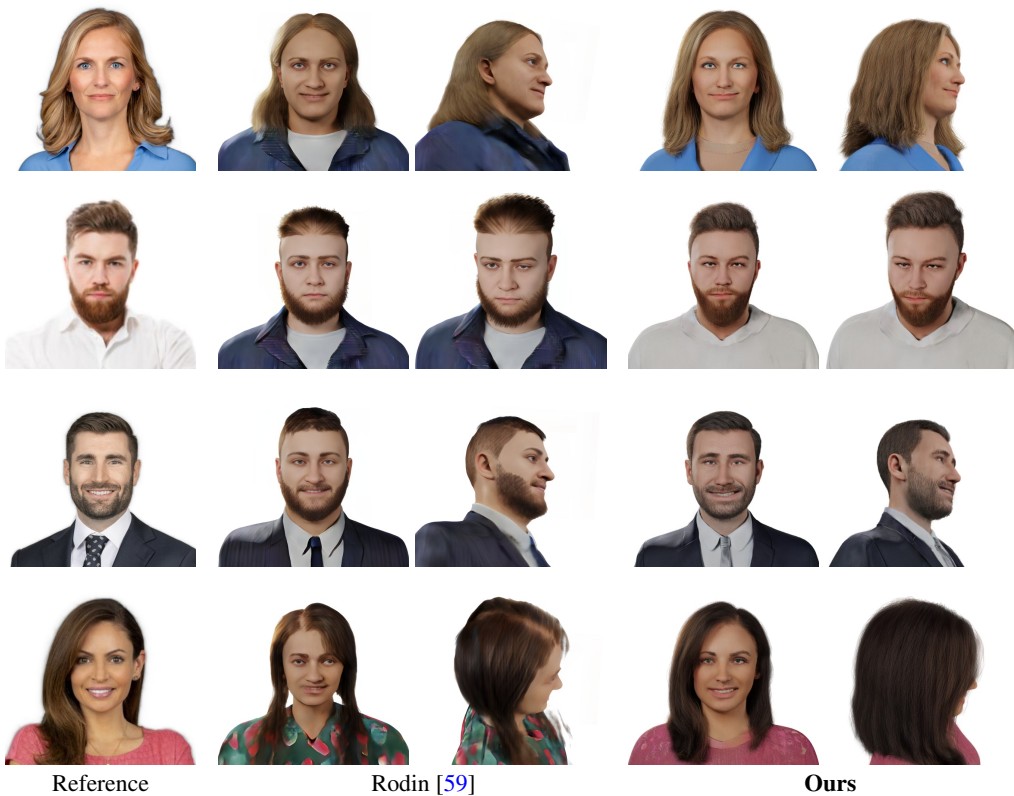

| Reference | Rodin [59] | **Ours** |

Figure 15: Additional qualitative comparison of 3D avatars creation conditioned on single in-the-wild portraits.

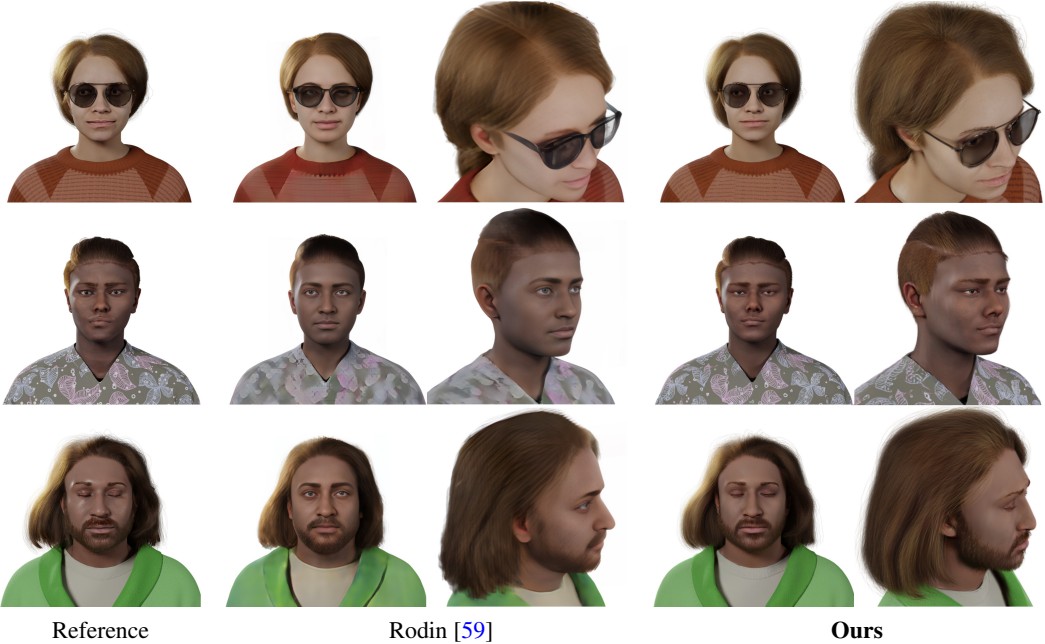

| Reference | Rodin [59] | **Ours** |

Figure 16: Qualitative comparison generated digital avatars conditioned on synthetic portraits.

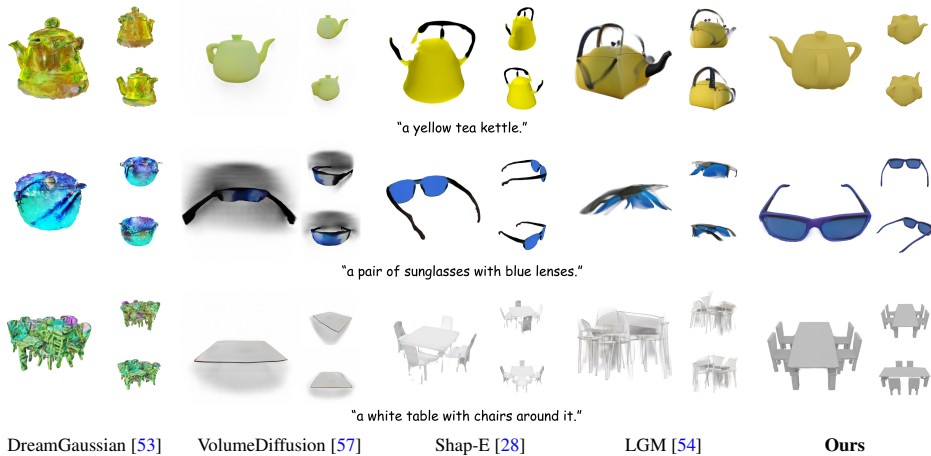

| DreamGaussian [53] | VolumeDiffusion [57] | Shap-E [28] | LGM [54] | **Ours** |

Figure 17: Additional qualitative comparison of text-to-3D generation on Objaverse [14]. Our model is capable of creating high-quality samples following input text prompts.

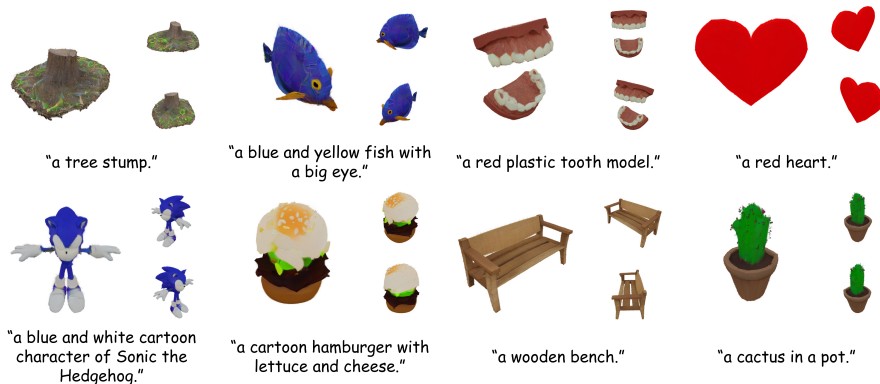

Figure 18: Additional results of text-to-3D generation.

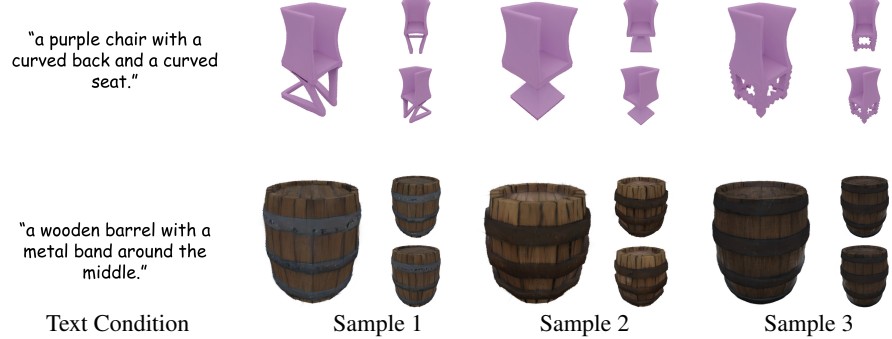

| Text Condition | Sample 1 | Sample 2 | Sample 3 |

Figure 19: Variation of text-to-3D generation. Our model is able to generate diverse results conditioned on the same text prompt.

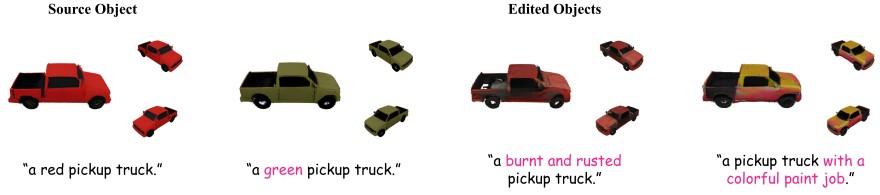

Figure 20: Example of text-guided 3D editing.

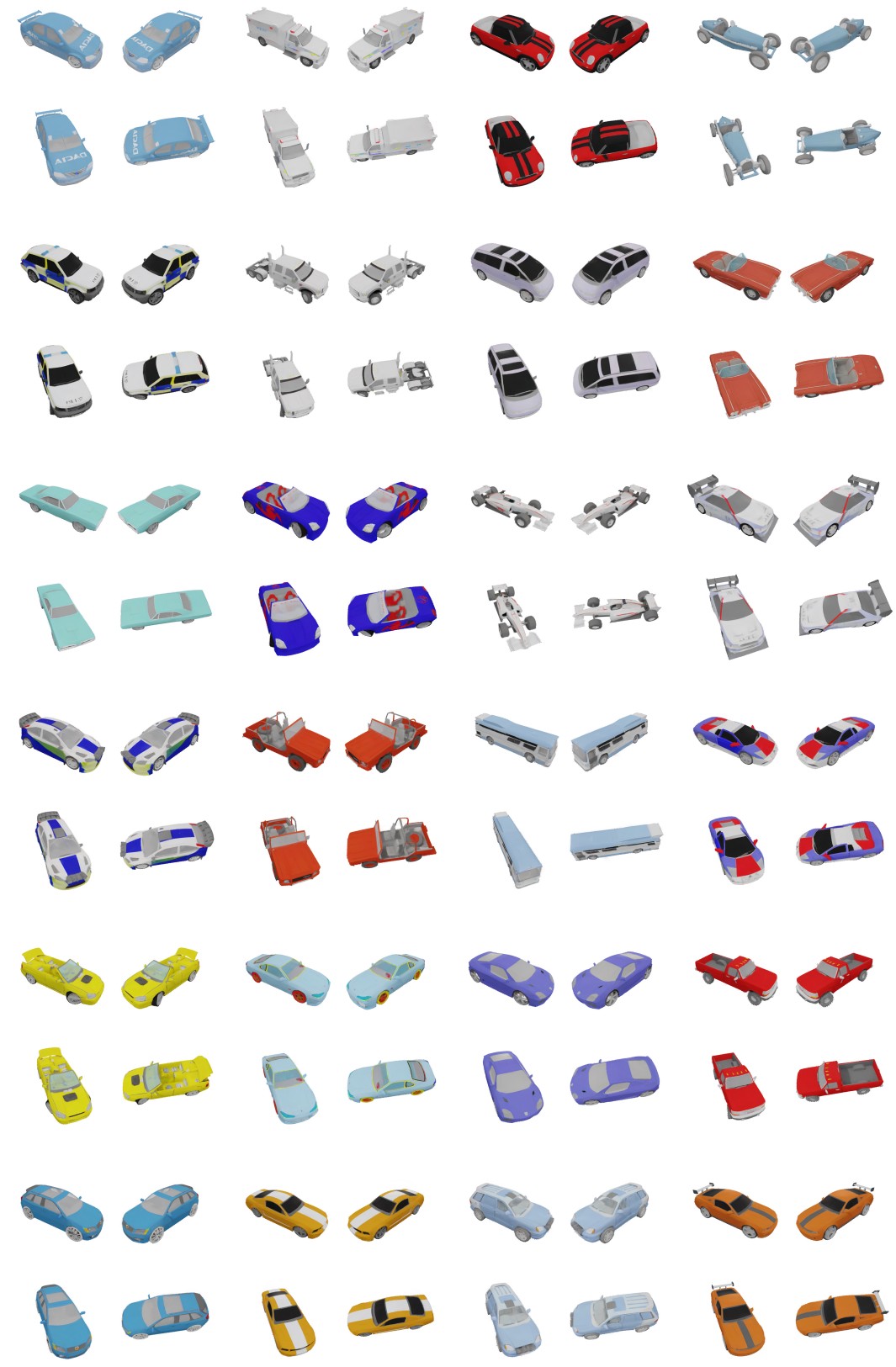

Figure 21: Additional generated samples on ShapeNet Car.

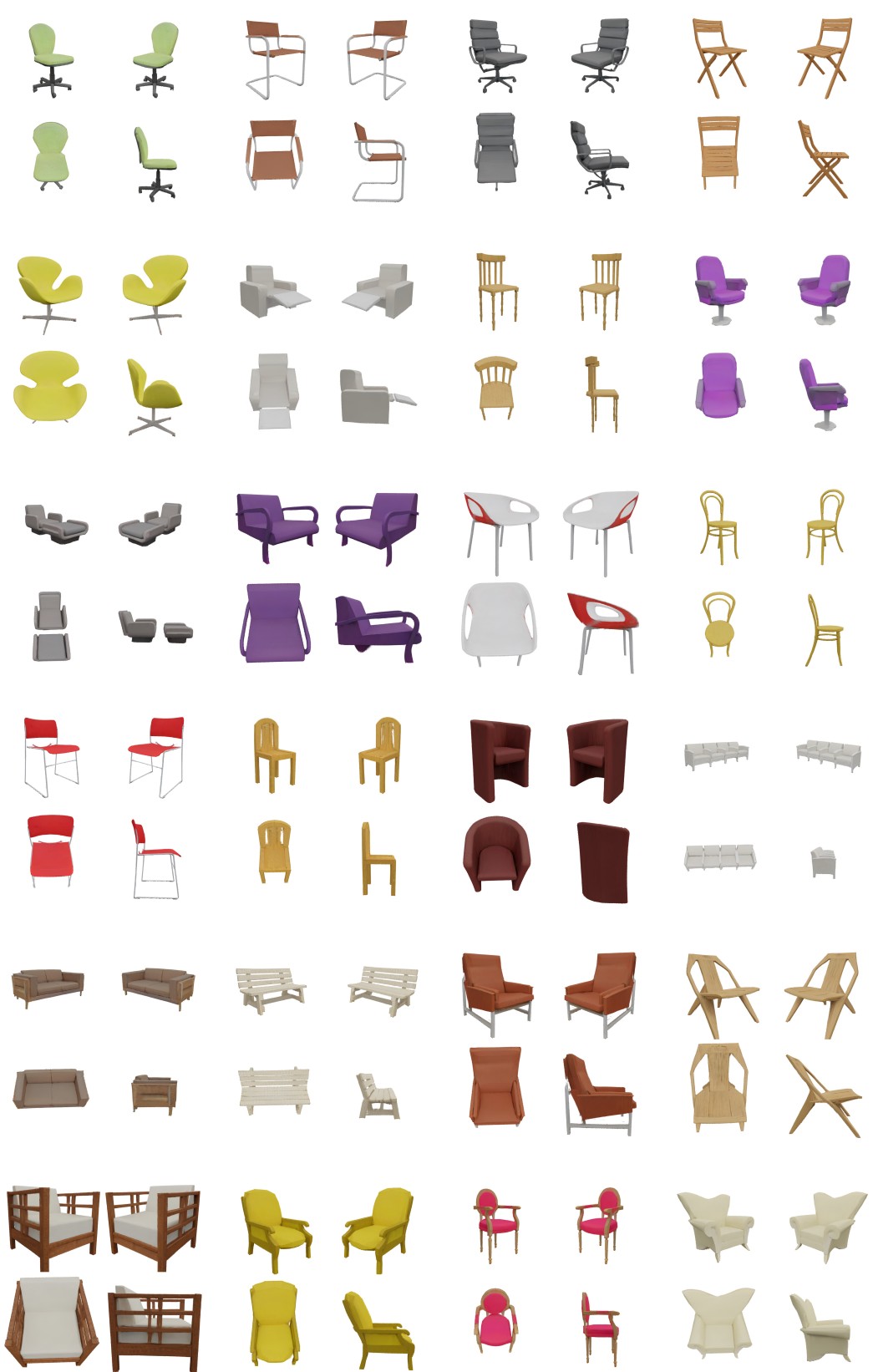

Figure 22: Additional generated samples on ShapeNet Chair.

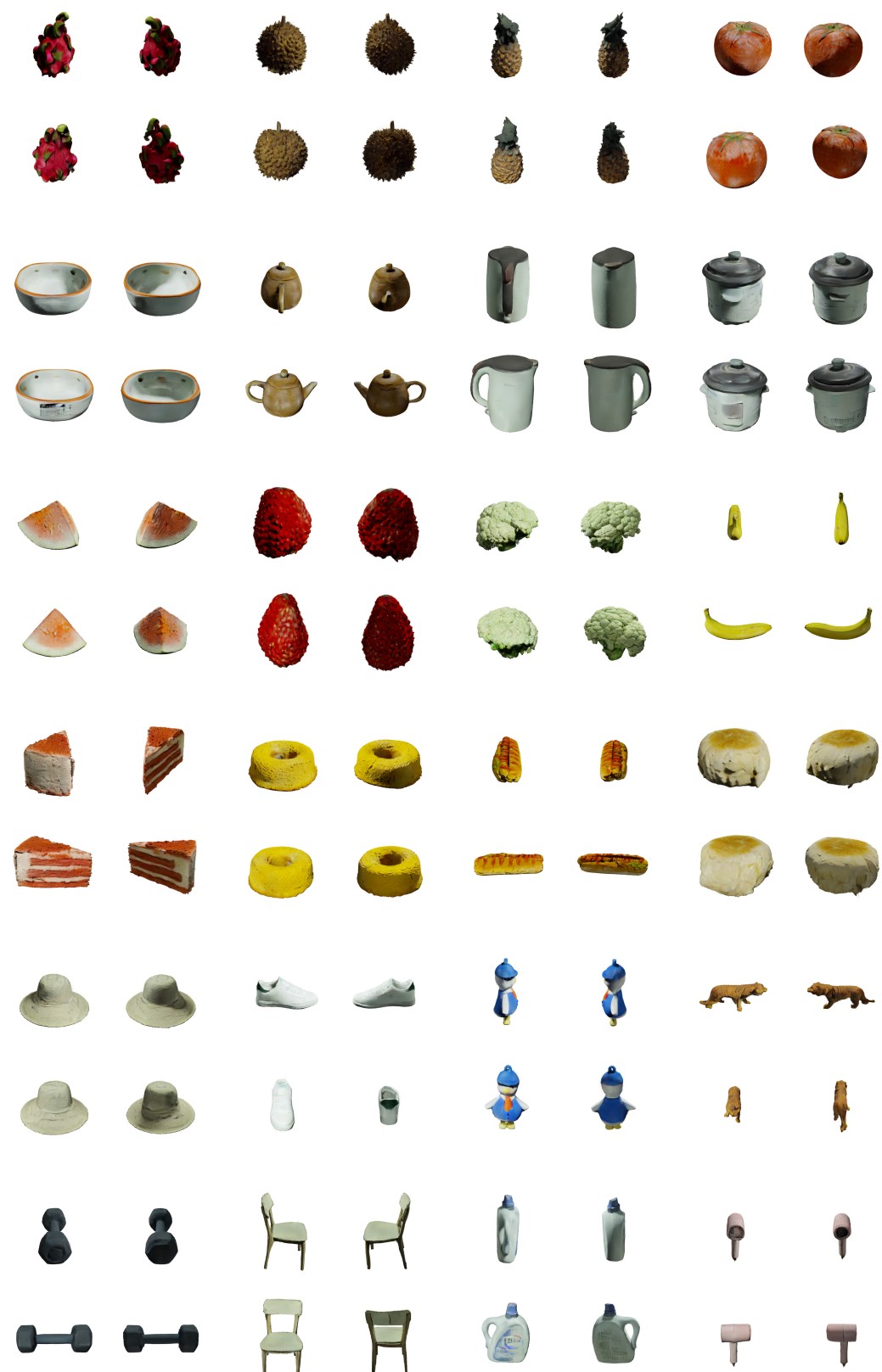

Figure 23: Additional generated samples on OmniObject3D.

