# OpenReview forum: "GaussianCube: A Structured and Explicit Radiance Representation for 3D Generative Modeling"
_NeurIPS.cc/2024/Conference — NeurIPS 2024 poster_

### Official Review · Reviewer_u6Hp · 2024-07-11

**Soundness:** 3
**Presentation:** 3
**Contribution:** 3
**Rating:** 7
**Confidence:** 5

**Summary:**

The paper introduces a novel explicit and structured 3D representation - GaussianCube - used in combination with U-Net diffusion models for 3D generation. While 3D Gaussians have a lot of advantages, they are not directly compatible with efficient architectures for generative modeling because of their unstructured nature. This paper tackles this problem by proposing two steps from 3D Gaussians to GaussianCube: 1) densification-constrained fitting to bound the number of Gaussians, and 2) structuralization via optimal transport to convert the underlying unstructured point cloud of 3D Gaussians to a structured voxel grid.

Using datasets in this format, the authors train diffusion models parameterized as 3D U-Nets for object, avatar, and text-to-3D generation.

**Strengths:**

* The paper introduces a simple but effective idea of bringing structure into unstructured 3D Gaussians.
  - This small change to the representation makes it directly compatible with standard architectures for diffusion models.
  - The use of optimal transport for voxelization is novel and elegant.
  - The representation might have potential to enable more efficient / optimized network architectures in other 3D tasks as well.
* The paper is presented well:
  - The main approach is clear.
  - The paper is well-written and easy to follow.
* The paper provides an extensive experimental evaluation validating their approach:
  - Generation with GaussianCube is evaluated for 4 tasks on synthetic as well as real data: unconditional generation on ShapeNet, class-conditional generation on OmniObject3D, image-conditioned avatar generation on Synthetic Avatar, and text-to-3D generation on Objaverse.
  - Quantitative and qualitative results show strong performance compared to baselines.
  - The supplementary material includes a video with more convincing qualitative results.
  - The ablation study validates the design choices.

**Weaknesses:**

* Lack of clarity:
  - The paper does not include any details about the diffusion process on the properties of 3D Gaussians (except of normalization with dataset statistics in the appendix):
    - 3D Gaussians have parameters with very different distributions making the diffusion process on them non-trivial.
    - As one example, if rotations are represented as quaternions with a diffusion process adding simply Gaussian noise, there is no guarantee about the final generation being a valid rotation.
    - Also the distribution of offsets from voxel centers would be insightful, especially across different objects/scenes with different level of detail.
  - The fact that positions are replaced by offsets of the voxel centers became clear rather late in the paper (ll. 153 ff.) and should be explained earlier.
  - The optimal transport illustration in Fig. 3 is not intuitive. Are there only two valid pixel corners on the diagonal?

* Missing related work and baseline:
  - The related work about diffusion models on 3D representations is missing [1] and [2].
  - The state-of-the-art comparison on ShapeNet misses an important baseline [1] with strong performance.

[1] Single-Stage Diffusion NeRF: A Unified Approach to 3D Generation and Reconstruction, ICCV 2023
[2] Neural Point Cloud Diffusion for Disentangled 3D Shape and Appearance Generation, CVPR 2024

**Questions:**

* Why do you use $x_0$-parameterization instead of $\epsilon$ or v?
* The resolution 32 is rather small and the appendix shows stronger fitting performance for larger numbers of Gaussians. Is there potential for latent diffusion?
* Why is the reported performance of DiffTF worse than the one from the original paper?

**Limitations:**

The authors addressed limitations and broader (including potential negative societal) impact in the appendix.

---

> ### Author Rebuttal · Authors · 2024-08-06
>
> Thank you for your valuable comments and suggestions. We address the reviewer's concerns below:
>
> > Q: Parameter distribution of 3D Gaussians used for diffusion process.
>
> A: Although the parameters of 3D Gaussians have very different distributions, we observe that after applying data normalization, they mainly obey the normal distribution, i.e., $\mathcal{N}(\mathbf{0}, \boldsymbol{I})$. Consequently, the data distribution does not present any significant divergence from that employed in the conventional diffusion process.
>
> > Q: How to guarantee the predicted properties are valid?
>
> A: We employ quaternion representations for rotations and normalize the predicted rotations to procure a valid unit quaternion following original Gaussian Splatting. We clamp the prediction of opacity $\alpha$ to $[0, 1)$ and clamp the minimum value of predicted scaling $\mathbf{s}$ to $0$ to ensure validity. No additional operations are applied to the positional and color feature predictions. Our findings suggest that these processes effectively validate the predicted Gaussians, yielding satisfactorily rendered images. We will incorporate these specifics into the revised manuscript.
>
> > Q: The distribution of the offsets from voxel centers.
>
> A: Thank you for your insightful suggestion. Please see Figure 3 in the attached PDF in the top-level comment. We visualize the offset distribution of 1K randomly selected GaussianCubes from each experimental dataset in Figure 3. We observe that most distributions exhibit a bell curve, similar to a normal distribution. However, the Digital Avatar dataset presents a more uniform distribution with multiple peaks. We believe these distributions offer valuable insights into how well the fitted 3D Gaussians align with voxel grid centers. Bell-shaped distributions akin to a normal distribution, such as in the ShapeNet Car and Chair datasets, suggest a strong initial alignment and lower complexity. On the other hand, broader distributions (e.g., the Digital Avatar dataset) indicate a higher level of detail (for instance, hair) and a greater need for adjustments during organization.
>
> > Q: The positions of Gaussians are replaced by offsets of voxel centers should be explained earlier.
>
> A: Thanks for your valuable comments. We will explain this operation earlier in the revised paper.
>
> > Q: Illustration of Fig. 3 of the main paper.
>
> A: We apologize for any confusion caused by the optimal transport illustration in Figure 3 of the main paper, where we provided a 2D toy example of two pixels. Actually, each pixel on the grid is valid rather than just two pixel corners on the diagonal. We appreciate your insightful suggestion and will revise this figure accordingly in the updated version.
>
> > Q: Missing related works of [1] and [2].
>
> A: Thanks for pointing out this. We will discuss these works in our revision.
>
> > Q: Missing comparison with [1] on ShapeNet.
>
> A: We have extended our analysis to include a comparison with SSDNeRF[1] on ShapeNet Car. By evaluating the official model checkpoint of SSDNeRF, we obtained the FID and KID scores between 50K generated renderings and 50K ground-truth renderings, aligning with the data in Table 3 of the main paper. The quantitative evaluation indicates that our method outperforms SSDNeRF in terms of both the FID and KID scores. The visual comparison in Figure 4 of the attached PDF in the top-level comment demonstrates that our model is able to generate high-fidelity results, surpassing SSDNeRF on the level of detail.
>
> | Method | FID-50K↓ | KID-50K(‰)↓ |
> | :--------- | :-----------: | :-------: |
> | SSDNeRF[1] | 23.10 | 11.68 |
> | Ours | **13.01** | **8.46** |
>
> > Q: Why do we use $x\_0$-parameterization instead of $\epsilon$ or $v$?
>
> A: Previous work [3] demonstrates that $x\_0$, $\epsilon$ and $v$ predictions only result in varied loss weights in diffusion training. We investigate both of them during earlier exploration and find $x\_0$-parameterization achieves the best performance. Therefore, we employ $x\_0$-parameterization in all subsequent experiments.
>
> > Q: Potential for latent diffusion?
>
> A: That's an insightful observation. Existing 2D latent diffusion models successfully achieve scale-up and exhibit amazing generative results. As such, 3D latent diffusion may potentially enhance the scalability of our model. However, this would necessitate the additional training of a Variational Autoencoder (VAE) that needs to be carefully designed to maintain the quality of reconstruction. We plan to explore this aspect further in our future research.
>
> > Q: Performance of DiffTF.
>
> A: We infer the official checkpoints of DiffTF and report the FID score between 50K generated and 50K ground-truth renderings rather than 50K generated and all ground-truth renderings used in the original paper. All reported FID and KID in Table 3 of the main paper are evaluated between 50K generated and 50K GT.
>
> [1] Chen H, Gu J, Chen A, et al. Single-stage diffusion nerf: A unified approach to 3d generation and reconstruction[C]//Proceedings of the IEEE/CVF international conference on computer vision. 2023: 2416-2425.
>
> [2] Schröppel P, Wewer C, Lenssen J E, et al. Neural Point Cloud Diffusion for Disentangled 3D Shape and Appearance Generation[C]//Proceedings of the IEEE/CVF Conference on Computer Vision and Pattern Recognition. 2024: 8785-8794.
>
> [3] Salimans T, Ho J. Progressive distillation for fast sampling of diffusion models[J]. arXiv preprint arXiv:2202.00512, 2022.

---

> ### Comment · Reviewer_u6Hp · 2024-08-10
> **Post-Rebuttal**
>
> I thank the authors for their clarifications regarding my concerns and providing additional results. The rebuttal addresses the mentioned weaknesses and questions. After carefully considering all reviews and answers by the authors, I would like to increase the rating to 7: Accept (see edited score in review).
>
> The following points prevent me from giving an even higher rating:
> - Limited contribution: Application of optimal transport for voxelization
> - Scalability concerns:
>   - high computational demand for training,
>   - rather low resolution of 32x32x32,
>   - and therefore the limitation to objects only

---

> > ### Author Response · Authors · 2024-08-13
> >
> > Dear Reviewer,
> >
> > We appreciate your thoughtful feedback and the increase in rating. We recognize that our current model demands high computation and we have only shown results of GaussianCube with a $32\times32\times32$ resolution. This is due to the inherent complexity of 3D generation. We're working hard to make our model more efficient. As we continue to learn and improve our techniques, we plan to apply our approach to more complex cases like 3D scene generation.
> >
> > Thank you again for raising the score.

---

### Official Review · Reviewer_Z6fD · 2024-07-12

**Soundness:** 3
**Presentation:** 3
**Contribution:** 3
**Rating:** 7
**Confidence:** 4

**Summary:**

The paper is about 3d object generative model. The generative objects are represented with gaussian splatting. Thus the main obstacle is the large point clouds. It is difficult to generate such large point clouds. The authors proposed a way to put point cloud to a structured grid using optimal transport. Some generative experiments are shown.

**Strengths:**

The idea of using optimal transport here is creative. And there is a motivation behind this.

The figures are also nice to illustrate the idea.

There are some kinds of experiments to verify the idea, including, objects and human avatars.

Generally I like the proposed idea and the convincing results shown in the main paper (and the video).

**Weaknesses:**

I would like to see some time and memory analysis about the method, including data preprocessing (object fitting and optimal transport), training resources, and sampling speed. These are important for readers to understand how difficult it is to reimplement the method.

Another thing might be useful is novelty analysis. Shapenet is a relatively small dataset, which can be easily memorized.

**Questions:**

The digital avatar experiment is image-conditioned generation? Is it possible to train an unconditional generative model?

I do not have further questions for this paper.

**Limitations:**

Yes.

---

> ### Author Rebuttal · Authors · 2024-08-06
>
> Thank you for your valuable comments and suggestions. We address the reviewer's concerns below:
>
> > Q: Time and memory analysis of our method.
>
> A: Thanks for your suggestions. As detailed in the supplementary material (Lines 501-503), the proposed densification-constrained fitting requires approximately 2.67 minutes on a single V100 GPU for each object over 30K iterations. The OT-based structuralization takes around 2 minutes per object on an AMD EPYC 7763v CPU. Fitting a single object consumes around 1GB of GPU memory. OT is run on CPU and costs around 5.4GB memory. For the diffusion training, we deploy 16 Tesla V100 GPUs for the ShapeNet Car, ShapeNet Chair, OmniObject3D, and Synthetic Avatar datasets, whereas 32 Tesla V100 GPUs are used for training on the Objaverse dataset. It takes about one week to train our model on ShapeNet Car, ShapeNet Chair, and OmniObject3D, and approximately two weeks for the Synthetic Avatar and Objaverse datasets. We provide the detailed inference configuration of DPM-solver and speed under various inference timesteps in the table below. Please note that all the inference times of the diffusion models in Table 5 of the main paper are reported using 100 steps to ensure a fair comparison.
>
> | Model size | Orders of Solver | Inference Mode | Inference Timesteps | Time (s) |
> |:------------:|:--------:|:-------------:|:-----------------:|:----------:|
> | 82M        | 3      | Multi-steps | 300             | 10.06    |
> |            | 3      | Multi-steps | 200             | 6.57     |
> |            | 3      | Multi-steps | 100             | 3.27     |
> | 339M       | 2      | Multi-steps | 300             | 15.54    |
> |            | 2      | Multi-steps | 250             | 13.80    |
> |            | 2      | Multi-steps | 200             | 10.64    |
> |            | 2      | Multi-steps | 100             | 5.13     |
> |            | 2      | Adaptive    | 44              | 2.30     |
>
> > Q: Novelty analysis of generation results on ShapeNet.
>
> A: In our experiments, we perform nearest neighbor search of our generated samples on ShapeNet dataset, as depicted in Figure 12 of the main paper. The results demonstrate that our model is capable of generating novel geometry and textures rather than simply memorizing the training data.
>
> > Q: Is the digital avatar creation model conditioned on image?
>
> A: Yes. The digital avatar creation model is conditioned on a single portrait image.
>
> > Q: Is it possible to train an unconditional digital avatar creation model?
>
> A: Certainly. We can train an unconditional model on a digital avatar dataset using the same methodology as with the ShapeNet datasets. In fact, our image-conditioned digital avatar creation diffusion model utilizes classifier-free guidance, allowing us to conduct unconditional inference simply by dropping the image conditions. Please refer to Figure 2 of the attached PDF in the top-level comment for our unconditionally generated digital avatars.

---

> ### Author Response · Authors · 2024-08-13
>
> Dear Reviewer,
>
> We have tried our best to address your questions as detailed in our top-level comment and the rebuttal above. Once again, we extend our gratitude for your time and effort in reviewing our manuscript, and we stand ready to provide any additional information that could be helpful.

---

### Official Review · Reviewer_E9Nw · 2024-07-13

**Soundness:** 4
**Presentation:** 3
**Contribution:** 3
**Rating:** 6
**Confidence:** 4

**Summary:**

The paper proposes an approach to 3D generation at the object level with the help of diffusion models. The main challenge in using diffusion model to generate 3D is the choice of 3D representation that fits well with the denoising network. The paper proposes using fixed number of Gaussians lying on a regular 3D grid. Each cell of the 3D grid stores the parameters of the Gaussians along with an offset from the cell. However, the Gaussians trained using Gaussians Splatting can lie on arbitrary coordinates. In order to assign a Gaussian to the cell of the grid, the paper proposes using Optimal Transport algorithm. This datastructure is called GaussianCube and result in shape being represented with smaller number of parameters.

Based on this GaussianCube data structure, we can train a 3D diffusion model to generate 3D shapes via Gaussians. This network can also be used to predict Gaussians in conditional manner e.g. image, text etc.

The resultant generative model produces 3D consistent representation via Gaussian splatting, follows conditioning signal accurately, fast inference (5 sec) and lower number of parameters.

**Strengths:**

The paper proposes a novel approach to consistent 3D generation with the help of diffusion model. The formulation of 3D dataset structure GaussianCubes is interesting as it allows 3D convolution on an unstructured data i.e. set of Gaussians.

The paper is mostly well written and describes the proposed approach well.

Experiments are performed on relevant dataset and evaulated against valid baselines.

Overall it is a good work showcasing how to combine diffusion model with Gaussian splatting in an end to end training resulting in better performance across several benchmarks.

**Weaknesses:**

1. Despite good results, my main concern is in the scalability of this approach. Given enough compute, we will be limited by the amount of 3D data. This is one of the reason of many efforts in 3D generative models based on images or videos only, as they are in abundance. I am skeptical can be used to generate assets for real applications. (Authors rebutting this should focus on how data scalability be achieved?)
2. How expressive and generalizable is the model in term of fitting many datasets (shapenet, objaverse etc) jointly? (Authors should not do experiments if they have not already tried training on all datasets.)
3. I find it quite surprising that the GaussianCube data structure works well even when trying to fit shapes with different scales and dimension. I expect that this representation will be hard to scale for complicated topology and intricate details where number of gaussians required to approximate the shape will be quite large. (Authors should discuss it in the rebuttal.)
4. Missing training time and compute requirement for training diffusion model. (Authors should provide this data in the paper.)

**Questions:**

1. In my opinion, Figure 3 is not required. The textual explanation is enough.
2. What are the failure cases of optimal transport? Have authors considered a learnable scattering of gaussians to cubic grid? (Authors should discuss this in the rebuttal.)

**Limitations:**

Limitations and societal impacts are described in the Supplementary material.

Though the scope of the paper is 3D asset generation and experiments are done accordingly, I would have appreciated some discussion on large scale scene generation also.

I suspect that there are failure cases of optimal transport. If so, they should be described.

---

> ### Author Rebuttal · Authors · 2024-08-06
>
> Thank you for your valuable comments and suggestions. We address the reviewer's concerns below:
>
> > Q: Given that many 3D generative models are based on images or videos due to the abundance of 2D data, how data scalability can be achieved for our approach?
>
> A: We acknowledge the reviewer's point that data holds significant importance in scaling up models. Despite that some 3D generation works are based on images and videos (e.g., MVDream, SV3D), a substantial amount of 3D data is still necessary for training in order to generate 3D consistent results. Therefore, we believe large-scale, high-quality 3D data is important for both our approach and the 2D-based methods. Recent trends indicate an active research interest in the development of expansive 3D datasets, such as Objaverse-XL[1], which boasts over 10 million 3D objects. The availability of such vast volumes of 3D data paves the way for us to scale up our model. Nevertheless, our approach is not limited to using 3D data only. We are also able to harness 2D image or video priors to further refine our generated results (e.g. leveraging Score Distillation Sampling with a pretrained 2D diffusion model).
>
> > Q: Expressiveness and generalization capability of our model to fit many datasets jointly.
>
> A: Regarding data fitting, the explicit nature of our representation eliminates the need for the shared implicit feature decoder and enables us to achieve high-quality fitting across all datasets. The fitting experiments in Table 1 and Table 2 of the attached PDF of the top-level comment demonstrate the expressiveness and generalization capability when jointly fitting different datasets (fitting results of GaussianCube are done before paper submission). Regarding modeling the distribution of GaussianCube using diffusion models, Objaverse serves as an exemplary case. This large-scale dataset with intricate data distribution encompasses most categories in ShapeNet. Given that our model yields high-quality generation results on Objaverse, we believe our method is effective and generalizable to handle complex datasets jointly.
>
> > Q: How to scale the representation for complicated topology and intricate details?
>
> A: GaussianCube is a highly expressive representation as we have demonstrated. A modest number of Gaussians (e.g., $32^3=32,768$) can effectively capture complex objects with high fidelity (e.g., the tire tread and hair of avatars in Figure 4 and Figure 7 of the main paper). Furthermore, we evaluate our method on the more challenging NeRF Synthetic dataset[2]. Our GaussianCube can approach the fitting quality of original GS using only $32,768$ Gaussians. For more complicated topology like Mic, our method achieves on-par quality by enlarging the corresponding voxel size and number of Gaussians (e.g., to $48^3=110,592$) without extra elaborate design to our pipeline. Please see the attached PDF in our top-level comment for visual comparison.
>
> | Number of Gaussians | Mic | Drums | Ficus | Ship  | Avg. |
> | :-------: | :------: | :-------: | :-------: |:-------: | :------: |
> | 32,768    | 33.25 | 25.76 | 34.55 | 29.90 |  30.87 |
> | 110,592   | 35.32 | 26.08 | 34.78 | 30.97 |  31.79 |
> | Original GS | 35.36 | 26.15 | 34.87 | 30.80 | 31.80 |
>
> > Q: Training time and compute requirement for diffusion model training.
>
> A: All of our diffusion model is trained using Nvidia Tesla V100 32G GPUs. It takes about one week to train our model on ShapeNet Car, ShapeNet Chair, and OmniObject3D, and approximately two weeks for the Synthetic Avatar datasets using 16 GPUs. For Objaverse dataset, the training duration extends to around two weeks with 32 GPUs. We will add the training time and compute information in the revised paper.
>
> > Q: Necessity of Figure 3.
>
> A: Thanks for your suggestion. We will consider revising it in the revision.
>
> > Q: Failure cases of optimal transport.
>
> A: We utilize the Jonker-Volgenant algorithm to resolve the optimal transport problem, which invariably yields the optimal solution. Consequently, we don't identify any failure cases of optimal transport in our model. If possible, we would appreciate further elaboration from the reviewer on what constitutes a failure case in the context of optimal transport. This would facilitate a more productive discussion during the reviewer-author dialogue period.
>
> > Q: Have we considered a learnable scattering of Gaussians to a cubic grid?
>
> A: That is indeed a compelling proposition. One potential advantage of a learnable scattering mechanism is the elimination of the time cost of OT. However, integrating it into the current pipeline presents several challenges. For example, the scattering process would need to be meticulously designed to ensure it is both differentiable and bijective. Additionally, jointly training the scattering with the diffusion model could potentially lead to instability issues. We will leave it to future research and thank the reviewer for the suggestion.
>
> > Q: Discussion of large-scale scene generation.
>
> A: Scene generation is an interesting topic. While the experiments conducted with GaussianCube primarily focus on objects, the generality of our pipeline suggests the potential applicability to scene generation as well. Nonetheless, scene generation comes with its own set of challenges. For one, handling unbounded scenes would require specialized design (e.g., contracting unbounded space into a ball in MipNeRF-360), given that GaussianCube necessitates a voxel grid within the bounding box. Additionally, the scarcity of large-scale scene data poses significant challenges for generative modeling. We plan to delve into these aspects in our future research.
>
> [1] Deitke M, Liu R, Wallingford M, et al. Objaverse-xl: A universe of 10m+ 3d objects[J]. Advances in Neural Information Processing Systems, 2024, 36.
>
> [2] Mildenhall B, Srinivasan P P, Tancik M, et al. Nerf: Representing scenes as neural radiance fields for view synthesis[J]. Communications of the ACM, 2021, 65(1): 99-106.

---

> ### Author Response · Authors · 2024-08-13
>
> Dear Reviewer,
>
> We have tried our best to address your questions as detailed in our top-level comment and the rebuttal above. Once again, we extend our gratitude for your time and effort in reviewing our manuscript, and we stand ready to provide any additional information that could be helpful.

---

### Official Review · Reviewer_hQzy · 2024-07-15

**Soundness:** 3
**Presentation:** 3
**Contribution:** 3
**Rating:** 6
**Confidence:** 4

**Summary:**

The paper proposes a new structured and explicit representation based on 3D Gaussian Splatting for 3D generation. The key idea is to properly organize the 3D Gaussians into a fixed-size volume, allowing for the use of the standard 3D UNet for diffusion. First, the paper uses a densification-constrained fitting algorithm to constrain the number of 3D Gaussians to be no more than a fixed value. Then, the paper proposes to use Optimal Transport to obtain a better spatial structure for the 3D Gaussians. The proposed representation can be easily used for both unconditional and conditional 3D generation. Experiments show that the proposed method outperforms baseline approaches.

**Strengths:**

Originality:

The paper proposes a new method to encode 3D Gaussians into a structured and explicit voxel grid, which is novel. The key novelty differs from the concurrent work [21]. While [21] also uses a voxel grid to organize 3D Gaussians, this paper proposes using Optimal Transport to make learning easier. This might inspire future research.

Quality:

The paper is technically sound. The author provides detailed implementation details.

Clarity:

The paper is mostly clear and easy to read.

Significance:

Applying diffusion models to 3D Gaussians for 3D generation is an important research problem, and the paper provides one feasible solution to this problem. This problem is non-trivial due to the irregularity of 3D Gaussians. The paper directly addresses this issue and proposes using OT for better spatial organization. Overall, I think this paper might inspire future research.

**Weaknesses:**

There are some concerns that need to be clarified:

- In Table 2, how is the number of the Triplane parameters computed? Why is it larger than Voxels? In literature, people use Triplane because it is more efficient than voxels. In addition, L11-L13: when encoding a large dataset, do Triplane/voxels (with shared implicit decoder) still have more parameters than GaussianCube?

- Is Equation (4) used to train the diffusion model? How do you obtain I_pred? (Do you run multiple denoising steps to generate it?) How does the expectation depend on t here?

**Questions:**

- The ShapeNet dataset is axis-aligned, but Objaverse is not. Objects in the Objaverse dataset might have different orientations. Is the method still able to generate meaningful results when the dataset size of Objaverse is small (e.g., only using ~5k cars from Objaverse)?

---

> ### Author Rebuttal · Authors · 2024-08-06
>
> Thank you for your valuable comments and suggestions. We address the reviewer's concerns below:
>
> > Q: Parameters of Triplane.
>
> A: In Table 2 of the main paper, we assigned the size of the Triplane as $3\times256\times256\times32$. We set the size of Voxels to $32\times32\times32\times14$ with the intent to draw a comparison with GaussianCube of a similar representation size. We additionally compare with Triplane of size $3\times128\times128\times10$, which has similar representation size with GaussianCube and Voxels of $32\times32\times32\times14$. Furthermore, we include results from Voxels of size $128\times128\times128\times32$, which yield much higher fitting quality than the $32\times32\times32\times14$ counterpart and exhibit a comparable fitting quality to both GaussianCube and Triplane. The additional results are included in the attached PDF of the top-level comment. As indicated in Table 1 of the attached PDF, the Triplane is considerably more efficient than the Voxels, offering superior fitting quality than Voxels with larger parameters. However, our GaussianCube still surpasses Triplane and Voxels in terms of fitting quality using the fewest parameters.
>
> > Q: When encoding a large dataset, do triplane or voxels (with shared implicit decoder) still have more parameters than GaussianCube?
>
> A: Yes. Due to the GaussianCube's explicit nature, it eliminates the requirement for a shared implicit decoder across various objects, leading to no significant difference in fitting quality when fitting larger datasets. Our representation size remains consistent at $32\times32\times32\times14$ across all datasets. We additionally conducted an experiment involving the fitting of the Objaverse dataset, which is not only larger but also exhibits a much more diverse distribution compared to ShapeNet. The experiment can be found in the attached PDF of the top-level comment, where we report the fitting quality of 100 randomly selected objects. As demonstrated in Table 2 of the attached PDF, our GaussianCube still achieves superior fitting quality while utilizing the minimum number of parameters compared with Voxels and Triplane.
>
> > Q: Is Equation (4) used to train the diffusion model?
>
> A: Yes, Equation (4) is used for diffusion training.
>
> > Q: How to obtain $I\_{\text{pred}}$?
>
> A: Our model is parameterized to predict the noise-free input $\mathbf{y}\_0$. For each training step, let $\hat{\mathbf{y}\_0} = \hat{\mathbf{y}\_\theta}\left(\alpha_t \mathbf{y}\_0+\sigma_t \mathbf{\epsilon}, t,  \mathbf{c}\_{\text{cls}}\right)$ be the prediction of our model. We can directly rasterize $\hat{\mathbf{y}\_0}$ to obtain $I\_{\text{pred}}$ without multiple denoising steps during diffusion training.
>
> > Q: How does the expectation depend on $t$ in Equation (4)?
>
> A: Given the model's prediction $\hat{\mathbf{y}\_0}$ and the camera parameters $\textbf{\textit{E}}$ used for rasteraization, we obtain $I\_{\text{pred}}$ by
> $$
> \begin{equation}
> \begin{array}{ll}
>     I\_{\text{pred}} &= \text{Rasterize}(\hat{\mathbf{y}\_0}, \textbf{\textit{E}}) \\\\
>     &= \text{Rasterize}(\hat{\mathbf{y}\_\theta}\left(\alpha\_t \mathbf{y}\_0+\sigma\_t \mathbf{\epsilon}, t,  \mathbf{c}\_{\text{cls}}\right), \textbf{\textit{E}}).
> \end{array}
> \end{equation}
> $$
>
> Therefore, Equation (4) can be written as
> $$
> \begin{equation}
> \begin{array}{ll}
>     \mathcal{L}\_{\text {image }} &= \mathbb{E}\_{t, I\_{\text{pred }}}\left(\sum\_l\left\\|\Psi^l\left(I\_{\text{pred}}\right)-\Psi^l\left(I\_{\text{gt}}\right)\right\\|\_2^2\right) +\mathbb{E}\_{t, I\_{\text {pred}}}\left(\left\\|I\_{\text {pred}}-I\_{\text {gt }}\right\\|\_2\right), \\\\
>     &= \mathbb{E}\_{t, \mathbf{y}\_0, \mathbf{\epsilon}}\left(\sum\_l\left\\|\Psi^l\left(\text{Rasterize}(\hat{\mathbf{y}\_\theta}\left(\alpha\_t \mathbf{y}\_0+\sigma\_t \mathbf{\epsilon}, t,  \mathbf{c}\_{\text{cls}}\right), \textbf{\textit{E}})\right)-\Psi^l\left(I\_{\text {gt}}\right)\right\\|\_2^2\right) \\\\
>     &+\mathbb{E}\_{t, \mathbf{y}\_0, \mathbf{\epsilon}}\left(\left\\|\text{Rasterize}(\hat{\mathbf{y}\_\theta}\left(\alpha\_t \mathbf{y}\_0+\sigma\_t \mathbf{\epsilon}, t,  \mathbf{c}\_{\text{cls}}\right), \textbf{\textit{E}})-I\_{\text {gt }}\right\\|\_2\right),
> \end{array}
> \end{equation}
> $$
>
> which illustrates that the expectation of Equation (4) depends on $t$. We thank the reviewers for this question and will further revise Equation (4) in the revision to improve its clarity.
>
> > Q: Is our method able to generate meaningful results on a small dataset containing objects of different orientations?
>
> A: That is an interesting question. Our method does not require the objects to be axis-aligned and has demonstrated strong capability to model complex distribution (e.g., Objaverse). Consequently, we believe our model is capable of providing meaningful results with a small dataset comprising objects of varying orientations. However, it is still challenging to fully represent the complete data manifolds using only a small amount of data samples. If the dataset size is excessively small, the model may be exposed to the risk of having limited extrapolation capabilities.

---

> ### Author Response · Authors · 2024-08-13
>
> Dear Reviewer,
>
> We have tried our best to address your questions as detailed in our top-level comment and the rebuttal above. Once again, we extend our gratitude for your time and effort in reviewing our manuscript, and we stand ready to provide any additional information that could be helpful.

---

### Author Rebuttal · Authors · 2024-08-06

Dear Reviewers,

We express our sincere gratitude to all reviewers for their valuable feedback, which has immensely contributed to the enhancement of our paper. We are greatly encouraged by the reviewers' acknowledgment that our paper:
- propose a novel method to address the irregularity of 3D Gaussians for 3D generation (Reviewer hQzy, Reviewer E9Nw), which might inspire future research (Reviewer hQzy).
- is creative and elegant to use Optimal Transport for Gaussian structuralization (Reviewer Z6fD, Reviewer u6Hp).
- is mostly well-written and clear and provides convincing results with strong performance (Reviewer hQzy, Reviewer E9Nw, Reviewer Z6fD, Reviewer u6Hp).

We have individually addressed each reviewer's concerns and also provided a one-page PDF incorporating additional tables and figures. If our responses adequately address your concerns, we would be grateful if you considered increasing your score. For any further questions, we are available for more detailed discussions.

In the attached one-page PDF, we additionally provide:
- Fitting comparison with Voxels of size $32\times32\times32\times14, 128\times128\times128\times32$ and Triplane of size $3\times128\times128\times10, 3\times256\times256\times32$ on ShapeNet Car (Table 1) and Objaverse (Table 2).
- Fitting results on NeRF Synthetic dataset using different numbers of Gaussians (Figure 1).
- Unconditionally generated samples of digital avatars by our approach (Figure 2).
- Distributions of offsets from voxel centers on each experimental dataset (Figure 3).
- Visual comparison with SSDNeRF on ShapeNet Car (Figure 4).

---

### Decision · Program_Chairs · 2024-09-25

**Decision:**

Accept (poster)

**Comment:**

The paper proposes a diffusion model for 3D Gaussians which is non-trivial due to the irregularity of 3D Gaussians. To use a standard 3D UNet for diffusion, the 3D Gaussians are confined into a fixed-size volume, and further optimized using Optimal Transport. The method outperforms the relevant baselines.

All reviewers are positive about the paper and the authors addressed some remaining concerns in their rebuttal. They point out that the presentation is good, the method appears technically sound and performs well. A shared concern is the scalability of the method, due to the need for 3D training data which the authors address in their rebuttal to some extent by reporting results for jointly fitting multiple datasets.

Overall, this is a solid paper and I recommend acceptance.